# Replication gap suppression depends on the double-strand DNA binding activity of BRCA2

Domagoj Vugic[1,2,5], Isaac Dumoulin[1,2,5], Charlotte Martin [1,2,5], Anna Minello [1,2], Lucia Alvaro-Aranda[3], Jesus Gomez-Escudero[3], Rady Chaaban[1,2,3], Rana Lebdy[4], Catharina von Nicolai[1,2], Virginie Boucherit[1,2], Cyril Ribeyre [4], Angelos Constantinou [4] & Aura Carreira [1,2,3] ✉

Replication stress (RS) is a major source of genomic instability and is intrinsic to cancer cells. RS is also the consequence of chemotherapeutic drugs for treating cancer. However, adaptation to RS is also a mechanism of resistance to chemotherapy. BRCA2 deficiency results in replication stress in human cells. BRCA2 protein's main functions include DNA repair by homologous recombination (HR) both at induced DNA double-strand breaks (DSB) and spontaneous replicative lesions. At stalled replication forks, BRCA2 protects the DNA from aberrant nucleolytic degradation and is thought to limit the appearance of ssDNA gaps by arresting replication and via post-replicative HR. However, whether and how BRCA2 acts to limit the formation of ssDNA gaps or mediate their repair, remains ill-defined. Here, we use breast cancer variants affecting different domains of BRCA2 to shed light on this function. We demonstrate that the N-terminal DNA binding domain (NTD), and specifically, its dsDNA binding activity, is required to prevent and repair/fill-in ssDNA gaps upon nucleotide depletion but not to limit PARPi-induced ssDNA gaps. Thus, these findings suggest that nucleotide depletion and PARPi trigger gaps via distinct mechanisms and that the NTD of BRCA2 prevents nucleotide depletion-induced ssDNA gaps.

Germline mono-allelic mutations in *BRCA2* predispose to breast and ovarian cancer with high penetrance[1]; when biallelic, they result in Fanconi anemia (FA)[2].

BRCA2 tumor suppressor protein preserves genomic integrity through its mediator role in DNA repair by homologous recombination (HR)[3–5]. On the one hand, BRCA2 loads and modulates the DNA binding activity of RAD51 preventing its non-productive association with double-stranded DNA (dsDNA) and promoting its nucleation onto the resected single-stranded DNA (ssDNA). On the other, it helps displace RPA from ssDNA thus facilitating RAD51 nucleoprotein filament formation and strand invasion. Replication stress induces the formation of DNA lesions that block replication; under these conditions, RAD51 protects stalled replication forks from unscheduled nucleolytic degradation, a function that is promoted by BRCA2[6–8]. In addition, RAD51 promotes replication fork reversal, a structure resulting from the annealing of the newly synthesized strands allowing to skip of a lesion[9,10] and restart of replication; these functions seem to be independent of BRCA2.

[1]Institut Curie, PSL Research University, CNRS, UMR3348, F-91405 Orsay, France. [2]Paris-Saclay University CNRS, UMR3348, F-91405 Orsay, France. [3]Genome Instability and Cancer Predisposition lab, Department of Genome Dynamics and Function, Centro de Biologia Molecular Severo Ochoa (CBMSO, CSIC-UAM), Madrid 28049, Spain. [4]Institut de Génétique Humaine, CNRS, Université de Montpellier, Montpellier, France. [5]These authors contributed equally: Domagoj Vugic, Isaac Dumoulin, Charlotte Martin. ✉e-mail: acarreira@cbm.csic.es

Replication stress leads to the appearance of stretches of ssDNA or single-strand DNA gaps (ssDNA gaps). These gaps have been shown to accumulate in BRCA1/2-deficient cells, especially under replication-compromising conditions, such as nucleotide depletion induced by hydroxyurea, after multiple rounds of cisplatin treatments, or upon treatment with PARP inhibitors[11] suggesting the involvement of these factors in preventing replication-associated ssDNA gaps[8,12–14]. The origin of ssDNA gaps is multiple varying from defects in Okazaki fragment processing to repriming by specialized polymerases[15]. Because they cannot be filled by conventional polymerases, several mechanisms need to act to "fill in" these ssDNA gaps left behind the forks. These include translesion synthesis (TLS), template switching (TS), and repriming by the primase-polymerase PRIMPOL (reviewed in refs. [15–17]). RAD51-mediated homologous recombination (HR) through TS can repair ssDNA gaps in an error-free manner[18] and has been shown to efficiently fill gaps opposite to bulky adducts in mammalian cells[19]. In the absence of a functional HR, such as in the BRCA1/2-deficient context, other mutagenic mechanisms take place to fill in the gaps including the TLS factors REV1-Polζ[20,21].

The fate of ssDNA gaps in BRCA2-deficient cells and how BRCA2 is involved in their suppression/repair is still poorly understood. The ortholog of BRCA2 in *U. maydis*, Brh2, can load Rad51 onto gapped DNA in vitro[22], an activity that requires a dsDNA/ssDNA junction[22], and this requirement was also shown for the loading of RecA by the functional homolog of BRCA2 in bacteria, RecFOR[23]. Although biochemical data is lacking, mammalian BRCA2 likely promotes ssDNA gap filled-in/ repair through an HR-dependent mechanism[8,19,24,25]. Finally, if left unrepaired, ssDNA gaps may persist or lead to replication-associated DSBs both of which can be repaired via HR in an error-free manner[21,26]. In the absence of a functional HR, these gaps or DSBs accumulate and manifest in chromatid gaps or breaks in metaphase spreads. The latter may be subjected to non-homologous end joining (NHEJ) repair that when ligated to different DSBs result in radial chromosomes both of which are observed in BRCA2-deficient cells[27].

Here, we investigate the role of BRCA2 at replication forks taking advantage of two variants in the N-terminal DNA binding domain (NTD) with either impaired dsDNA binding activity[27] or impaired ssDNA and dsDNA binding activities[28]. We find that the NTD of BRCA2 and in particular, its dsDNA binding activity, is required for ssDNA gap suppression. ssDNA gaps form in cells expressing BRCA2-C315S despite a functional fork arrest and persist through mitosis as detected in metaphase spreads in the same cell cycle. Consistently, cells bearing the NTD variants show hypersensitivity to replication stress induced by hydroxyurea (HU). In contrast, these cells are resistant to PARP inhibitors (PARPi) and do not accumulate ssDNA gaps in these conditions. These findings suggest that nucleotide depletion and PARPi trigger gaps in a different manner and therefore require distinct functions for their repair. Moreover, using a gene-targeting cell-based assay, we show that cells bearing NTD variants are proficient in DSB repair, suggesting that the dsDNA binding activity of BRCA2, located at the NTD, is necessary for the repair of replication-associated gaps but dispensable for the repair of DSBs. Reconstituting ssDNA gap repair in vitro, we find that RAD51 can perform recombination from an ssDNA gap mimicking substrate without the requirement of an ssDNA 3'-end and that BRCA2$_{NTD}$ can readily promote this reaction. Thus, the dsDNA binding activity of BRCA2 promotes ssDNA gap repair by HR.

Our findings establish BRCA2 dsDNA binding activity, unique to the NTD and impaired in the breast cancer variants C315S and S273L, as essential for the ssDNA gap suppression activity of BRCA2. These variants uncouple the function of BRCA2 in the recombinational repair of replication-associated gaps from the repair of DSBs.

## Results

### Breast cancer variants affecting the NTD or the CTD of BRCA2 confer different sensitivity to replication stress

In addition to the canonical C-terminus DNA binding domain (CTD), BRCA2 presents a second DNA binding site in the N-terminus (NTD)[28]. Unlike the CTD, which binds ssDNA[29], the NTD can bind both ssDNA and dsDNA in vitro[28]. The NTD can promote RAD51-dependent HR in isolation in vitro; however, the interdependencies between the NTD and the CTD in the context of the full-length BRCA2 and in cells have not been elucidated. Moreover, given the role of BRCA2 and HR in DSB repair and in the response to replication stress, we wondered which domains of BRCA2 were required for these different functions. To gain insight into these questions, we used breast cancer variants that impair the DNA binding ability of the NTD or the CTD. We focused on three variants, R3052W, a pathogenic variant that affects the ssDNA binding activity of the CTD and is therefore deficient in HR[30]. C315S, a variant of unknown clinical significance (VUS) located in the NTD that impairs specifically the dsDNA binding activity of BRCA2 and cannot promote RAD51-mediated recombination at a resected-DNA mimicking substrate[28]. Finally, we included a previously uncharacterized VUS located as well in the NTD region, S273L.

We have shown that the dsDNA binding activity of BRCA2 is located at the NTD and that the CTD is devoid of this activity[28]; however, these experiments were performed with a short dsDNA substrate. As it has been suggested that the Tower domain of the CTD may interact with dsDNA[29] we first compared the dsDNA binding activity of recombinant CTD (aa 2474–3190) and NTD (aa 250–500) this time using a longer $^{32}$P-labeled (191 bp) dsDNA substrate in electrophoretic mobility shift assays (EMSA) in vitro. Using this set-up, only the NTD showed dsDNA binding activity, confirming and extending our previous findings and indicating that the dsDNA binding activity of BRCA2 is unique to the NTD (Fig. 1a). We then purified the NTD bearing the VUS S273L and tested its DNA binding ability by EMSA with synthetic oligonucleotide substrates as previously performed for BRCA2$_{NTD-C315S}$[28]. BRCA2$_{NTD-S273L}$ impaired both ssDNA and dsDNA binding in vitro (Fig. 1b, Supplementary Fig. 1a, b).

We generated stable cell lines bearing the selected variants by transfecting the BRCA2 cDNA coding a GFP-MBP-tagged version of BRCA2 full-length protein (BRCA2 WT) or the variants C315S, S273L, or R3052W to complement DLD1 BRCA2 deficient human cells (hereafter BRCA2$^{-/-}$). In this cell line, both alleles of BRCA2 contain a deletion in exon 11 causing a premature stop codon after BRC5 and cytoplasmic localization of a truncated form of the protein[31]. When possible, we selected two stable clones of each variant to account for possible phenotypic differences observed due to the different protein levels among the clones compared to the wt clone (clone C1, hereafter BRCA2 WT) (Supplementary Fig. 2a). Next, we assessed the sensitivity of these cells to different genotoxic agents. We first tested their sensitivity to the poly (ADP-ribose) polymerase (PARP) inhibitor Olaparib. PARP1 is an enzyme required for the sensing of DNA single-strand breaks (SSBs) and double-strand breaks (DSBs) that becomes essential in the absence of a functional HR pathway[32–34]. PARP1 inhibitors, in particular Olaparib, are currently used in the clinic to treat breast and ovarian cancer patients carrying germline mutations in BRCA1/2[35]. In our settings, the relative viability of BRCA2$^{-/-}$ cells was 12% upon a 6-days treatment with the highest Olaparib concentration used (2.5 μM); in contrast, 70% of BRCA2 WT complemented cells remained viable. Similarly, cells expressing S273L or C315S survived the treatment to the same level as the cells expressing BRCA2 WT (Fig. 1c). Consistent with previous results[30], R3052W, the CTD variant that impairs HR activity, showed hypersensitivity to the treatment with only 30% of surviving cells at 2.5 μM Olaparib (Fig. 1c).

To directly assess the capacity of these cells to repair DSBs by HR we performed a cell-based HR assay. Based on the classical

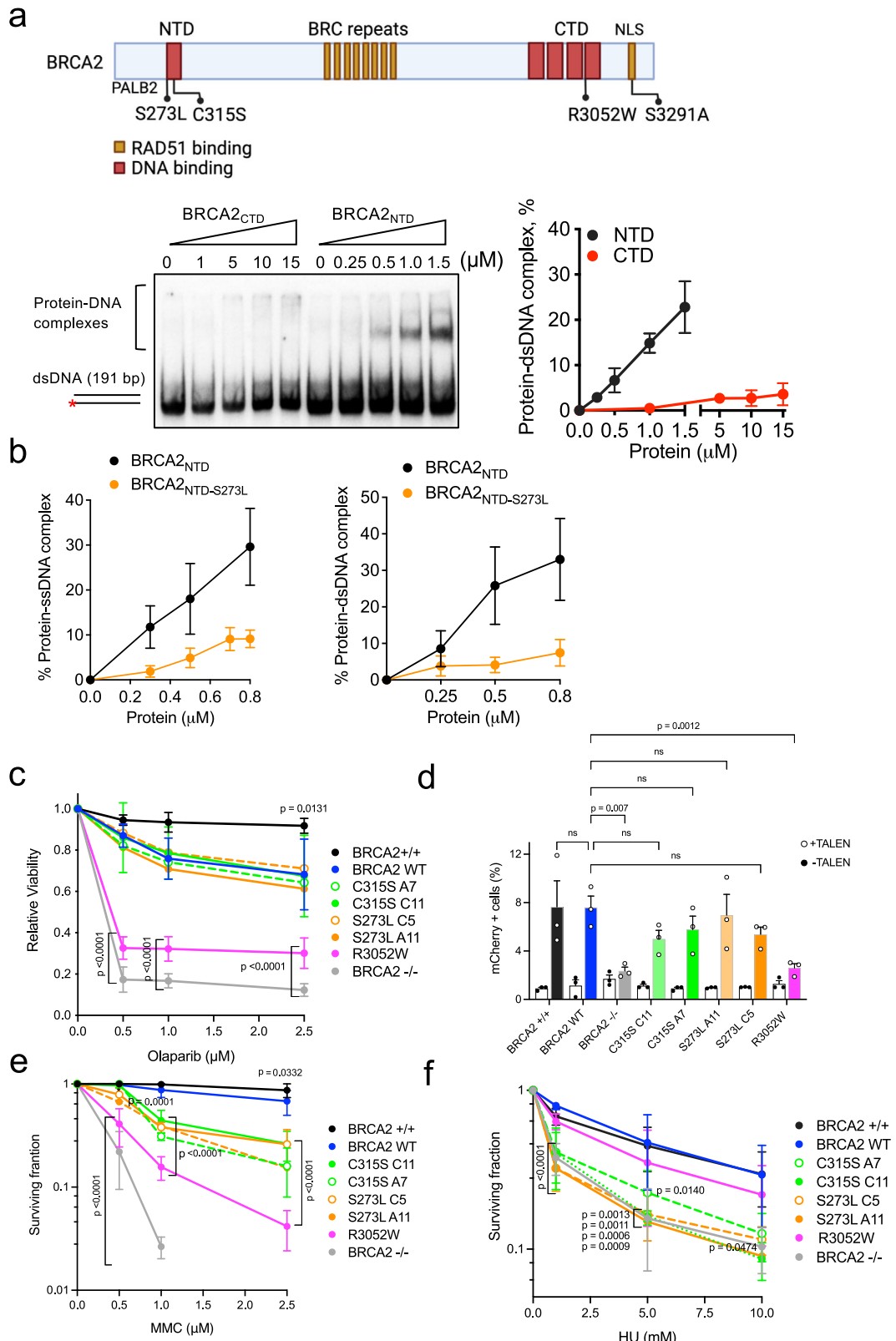

GFP-reporter assay[36], this test measures DSB-mediated gene targeting activity at a specific locus (AAVS1 site) within the endogenous PPP1R12C gene using a site-specific transcription-activator like effector nuclease (TALEN) and a promoter-less mCherry donor flanked by homology sequence to the targeted locus[37]. DSB-mediated gene targeting results in mCherry expression from the endogenous PPP1R12C

promoter which can be measured by flow cytometry. Using this system, cells expressing the endogenous BRCA2 protein (BRCA2[+/+]) or BRCA2 WT complemented cells showed ~7.5% of mCherry positive TALEN-transfected cells (mean of 7.6% for both BRCA2[+/+] and BRCA2 WT) whereas BRCA2-deficient cells (BRCA2[−/−]) showed only 2.3% of mCherry expressing cells, as expected (Fig. 1d, Supplementary Fig. 2b).

**Fig. 1 | BRCA2 variants located in the NTD are sensitive to replication stress.**
**a** (Top) Schematic representation of BRCA2 structure indicating its DNA binding domains (in red) and RAD51 binding sites (in orange) with the variants/mutations located within them used in this study. The location of the PALB2 binding site and the nuclear localization signal (NLS) are also indicated. Figure created with BioRender.com. (Bottom) Representative EMSA and quantification comparing the binding of increasing concentrations of BRCA2$_{NTD}$ and BRCA2$_{CTD}$, as indicated, to dsDNA (191 bp). The data represent the mean from three independent experiments. Error bars, SD. **b** Quantification of EMSA showing the binding of purified BRCA2$_{NTD}$ or BRCA2$_{NTD-S273L}$ at the indicated concentrations to ssDNA (dT$_{40}$) or dsDNA (42mer) $^{32}$P-labeled substrates. The data represent the mean from three independent experiments. Error bars, SD. See also Supplementary Fig. 1. **c** Quantification of the relative cell viability monitored by MTT assay upon treatment with increasing doses of the PARP inhibitor Olaparib, as indicated. The data represent the mean ± SD of four independent experiments. (ns, not significant). **d** Frequency of mCherry positive cells in cells transfected with the promoter-less donor plasmid (AAVS1-2A-mCherry) without (−TALEN) (open circles) or with (+TALEN) nucleases (filled circles). The error bars represent the mean ± SEM of three independent experiments. See also Supplementary Fig. 2b. **e** Quantification of the surviving fraction of BRCA2$^{+/+}$ and BRCA2$^{-/-}$ or BRCA2$^{-/-}$ stable clones expressing BRCA2 WT or the variants C315S, S73L, R3052W, assessed by clonogenic survival upon exposure to MMC at the indicated concentrations. Data represent the mean ± SD of three independent experiments. **f** Quantification of the surviving fraction of BRCA2$^{+/+}$ and BRCA2$^{-/-}$ or stable clones expressing BRCA2 WT or the variants C315S, S273L, R3052W, assessed by clonogenic survival upon exposure to HU at the indicated concentrations. Data represent the mean ± SD of three independent experiments. See also Supplementary Fig. 3. Statistical difference in **c–f** was determined by a two-way ANOVA test with Dunnett's multiple comparisons tests. The p-values show significant differences compared to the BRCA2 WT clone. Only significant p-values are shown. Source data are provided as a Source Data file.

**Table 1 | Main phenotypes observed in the different DLD1 BRCA2-mutated stable cell lines used in this study**

| Cell line | HR | FP | FA | GS HU | GS PARPi | PARPi response | HU response | MMC response |
|---|---|---|---|---|---|---|---|---|
| BRCA2 WT | + | + | + | + | + | Resistant | Resistant | Resistant |
| BRCA2$^{-/-}$ | − | − | − | − | − | Sensitive | Sensitive | Sensitive |
| S273L | + | / | − | − | / | Resistant | Sensitive | mild sens. |
| C315S | + | +/− | + | − | + | Resistant | Sensitive | mild sens. |
| R3052W | − | +/− | + | + | − | Sensitive | Resistant | Sensitive |
| S3291A | / | +/− | + | + | / | / | / | / |

*HR* homologous recombination, *FP* fork protection, *FA* fork arrest, *GS* gap suppression, *mild sens.* mild sensitivity.

In agreement with our previous report[30] cells expressing the CTD variant, R3052W, was HR deficient showing 2.6% of mCherry-expressing cells. Importantly, TALEN-transfected cells expressing BRCA2 variants S273L and C315S showed no significant difference with the BRCA2 WT complemented cells (means ranging from 5% to 7% depending on the clone) indicating a nearly normal or intact DSB repair activity by HR.

As a member of the Fanconi anemia pathway, BRCA2 (FANCD1)-deficient cells are extremely sensitive to crosslinking agents and platinum drugs such as cisplatin or mitomycin C (MMC)[2]; therefore, we next performed clonogenic survival assays to assess the sensitivity of cells bearing NTD variants and the CTD variant to increasing concentrations of MMC. As expected, BRCA2-deficient cells (BRCA2$^{-/-}$) showed hypersensitivity to MMC treatment already at 1 μM MMC whereas BRCA2 WT cells complemented this phenotype almost to the same survival levels as the cells expressing the endogenous BRCA2 (BRCA2$^{+/+}$) (Fig. 1e, Supplementary Fig. 3). Cells bearing R3052W, the HR-deficient variant, displayed hypersensitivity to MMC. The stable clones expressing variants S273L and C315S also showed increased sensitivity to MMC although they resulted in an intermediate phenotype between the BRCA2 WT cells and cells expressing R3052W (Fig. 1e, Supplementary Fig. 3). Given that MMC treatment primarily generates inter-strand crosslinks which can inhibit transcription and replication in addition to prompting DNA breaks[38]; we then tested whether the NTD or CTD variants rendered cells sensitive to other forms of replication stress. We exposed cells to hydroxyurea (HU), a drug that reduces the pool of dNTPs leading to stalled replication forks, and assessed their viability via clonogenic survival assay. DLD1 BRCA2-deficient cells were moderately sensitive to HU as compared to MMC treatment. Remarkably, the CTD variant R3052W restored the sensitivity to HU almost to BRCA2 WT levels. In contrast, BRCA2 NTD variants S273L and C315S displayed similar sensitivity to HU as the BRCA2-deficient cells (BRCA2$^{-/-}$) (Fig. 1f, Supplementary Fig. 3).

In conclusion, in the context of the full-length protein and in cells, the pathogenic mutation R3052W altering the DNA binding activity of CTD renders cells sensitive to PARPi and MMC but not HU, whereas the VUS altering the DNA binding activity of the NTD (either dsDNA binding or both ssDNA and dsDNA binding) conferred moderate sensitivity to MMC and high sensitivity to HU, comparable to the BRCA2-deficient cells (Table 1).

## BRCA2 co-localizes with nascent DNA and NTD, CTD variants and C-terminal mutant S3291A delay its recruitment to the fork

Given that the NTD variants confer HU sensitivity but their DSB repair activity appeared intact, we then tested whether BRCA2 was recruited to nascent DNA and whether or not the NTD variants altered this localization. To do so, we used a combination of click-chemistry with the thymidine analog EdU and in situ proximity ligation assay (PLA) to measure the association of proteins with nascent DNA[39]. In this assay, the stable cell lines bearing BRCA2 WT or BRCA2 mutated forms are labeled with EdU, biotin is then conjugated to the EdU by click chemistry and PLA is used to detect BRCA2 in proximity to biotin-labeled nascent DNA. Consistent with previous results in *X. laevis*[12], we found that BRCA2 was in proximity to nascent DNA during unperturbed replication indicated by the presence of PLA foci (Fig. 2a). Upon a low dose of HU (0.2 mM) which is however sufficient to stall replication forks[40,41], the levels of BRCA2 WT associated with nascent DNA increased ~2-fold at 1 h treatment. To find out whether BRCA2 was specifically associated with nascent DNA we used thymidine chase experiments as previously described[41] using the same set-up. As expected, the levels of PLA foci specific for histone H1-EdU were not altered at any time point following thymidine chase whereas the levels of PLA foci specific for PCNA-EdU, a protein that travels with the replication fork, were strongly reduced (Supplementary Fig. 4a). Thymidine chase strongly reduced the PLA signal of BRCA2-EdU in both unperturbed replication and replication stress conditions suggesting that at a large fraction of the BRCA2 pool is associated specifically with nascent DNA (Fig. 2b).

Similar to BRCA2 WT, BRCA2-R3052W was associated with nascent DNA in both conditions although the overall PLA signal was

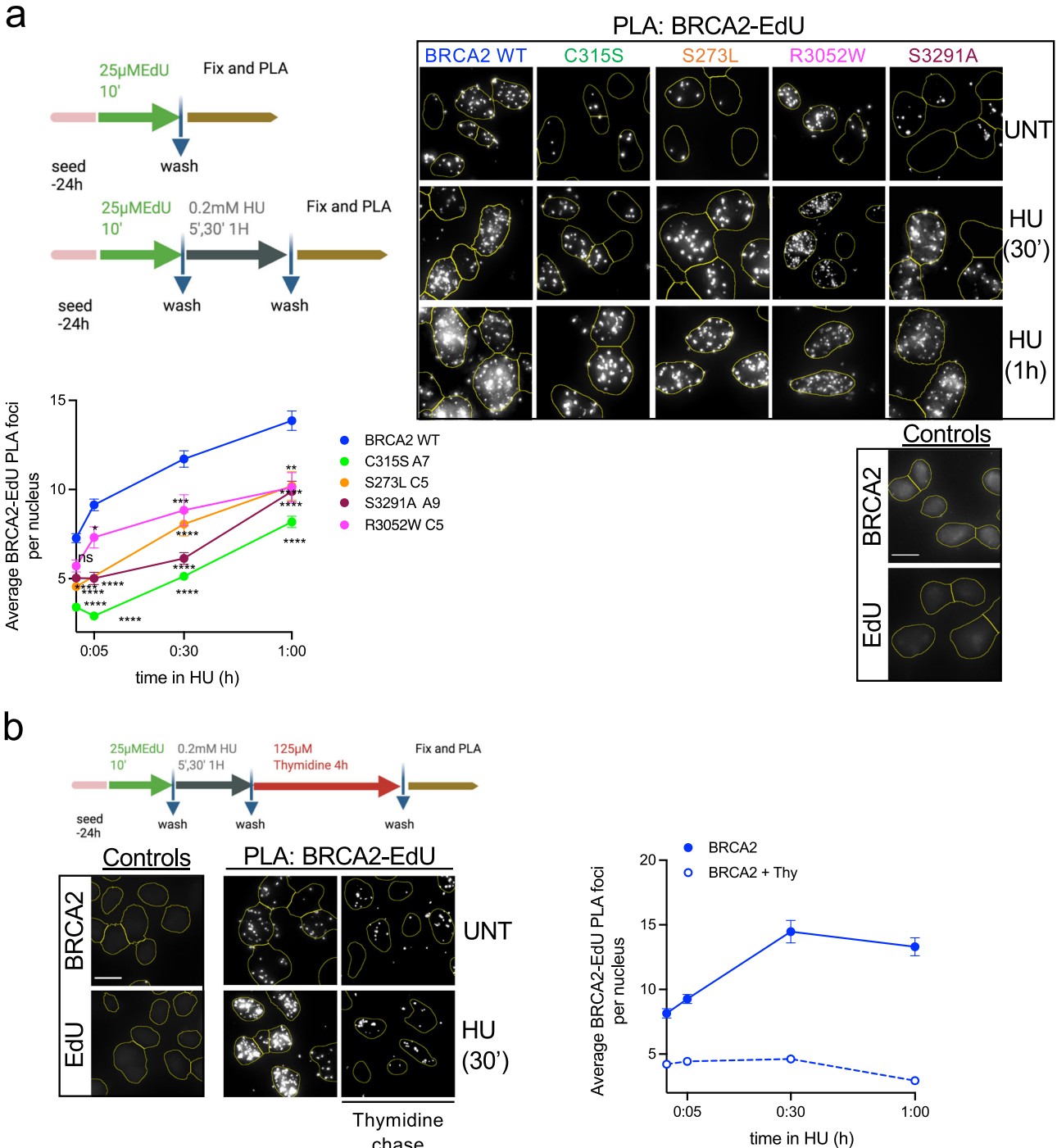

**Fig. 2 | BRCA2 NTD and C-terminal domains contribute to BRCA2 location at nascent DNA. a** (Top left) Scheme of the assay. (Top right): Representative images of in situ PLA on nascent DNA between biotinylated EdU detected with anti-biotin antibody and BRCA2-specific antibody in DLD1 BRCA2-deficient cells (BRCA2⁻/⁻) stably expressing either BRCA2 WT or the variants C315S (A7), S273L (C5), R3052W, and S3291A mutant, as indicated. Cells left untreated (UNT) or treated with HU (30' or 1 h 0.2 mM) are shown. An individual signal is observed (focus) if the two probed proteins (BRCA2 and EdU-Biotin) are in close proximity (<40 nm). For all the experiments we carried out two single-antibody control (anti-BRCA2 and anti-biotin) to assess the specificity of the PLA signal. The scale bar indicates 10 μm. (Bottom) Quantification of the BRCA2 recruitment was measured as the number of PLA foci observed per nucleus. The data represent the mean + SEM with 200–300 cells analyzed in each experimental data set at each time point. The number of independent experiments performed was as follows: BRCA2 WT: (UNT $n = 10$; 5′ $n = 5$; 30′ $n = 8$; 1 h $n = 7$), C315S: (UNT $n = 6$; 5′ $n = 3$; 30′ $n = 5$; 1 h $n = 5$); S273L:(UNT $n = 4$; 30′ $n = 2$; 1 h $n = 2$); S3291A: (UNT $n = 3$; 5′ $n = 2$; 30′ $n = 2$; 1 h $n = 2$); R3052W:

(UNT $n = 6$; 5′ $n = 3$; 30′ $n = 3$; 1 h $n = 3$). Statistical difference was determined by the Kruskal−Wallis test followed by Dunn's multiple comparison test. The $p$-values show significant differences compared to the BRCA2 WT clone. ns, not significant ($p = 0.7539$ in UNT BRCA2 WT vs. R3052W), *$p = $ <0.05 ($p = 0.0136$ at 5′ BRCA2WT vs. R3052W), **$p < 0.01$ ($p = 0,0042$ at 1 h BRCA2WT vs. R3052W), ***$p < 0.001$ ($p = 0.0002$ at 30′ BRCA2WT vs. R3052W), ****$p < 0.0001$). **b** (Left) Scheme of the assay and representative images of in situ PLA on nascent DNA between biotinylated EdU detected with anti-biotin antibody and BRCA2-specific antibody in DLD1 BRCA2 WT cells after 4 h Thymidine chase in cells left untreated or treated with HU (30′ 0.2 mM). (Right) Quantification of the BRCA2 recruitment measured as the number of PLA foci observed per nucleus after 4 h Thymidine chase in BRCA2 WT cells at different time points. The data represent the mean + SEM of two independent experiments with 200–300 cells analyzed in each experimental data set at each time point. Schemes of the PLA assay created with BioRender.com. Source data are provided as a Source Data file.

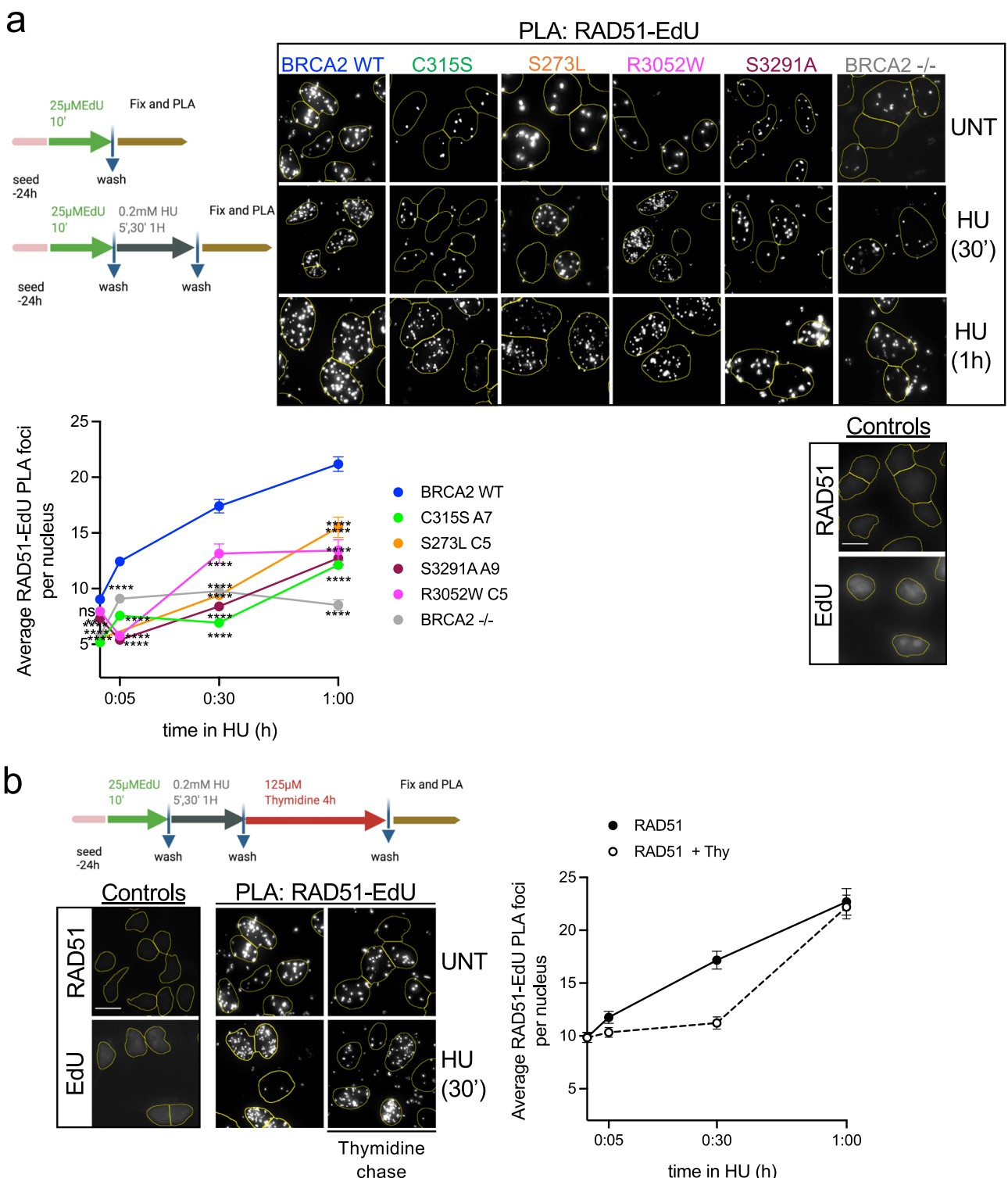

reduced (Fig. 2a). Next, we generated cell lines expressing a mutant of BRCA2 located at the extreme C-terminus of BRCA2, S3291A (Supplementary Fig. 4b), which reduces RAD51 oligomer binding[42] and was previously shown to impair replication fork protection[7]. Although present at the fork, the overall proximity of BRCA2-S3291A mutant to nascent DNA was considerably reduced (2-fold reduction after 30 min of HU treatment). A similar trend was observed for cells bearing BRCA2-S273L. Interestingly, cells expressing BRCA2-C315S showed an ever further reduction at all time points compared to the other cell lines (Fig. 2a).

The cell growth differences between the clones could have an impact on the levels of EdU incorporation; therefore, we controlled for the replication levels by performing biotin–biotin PLA[43]. No significant difference was observed in the EdU PLA signal in cells expressing BRCA2 WT compared to C315S, S273L, R3052W, or S3291A (Supplementary Fig. 4c). Importantly, we performed this assay in stable clones that show similar or higher levels of BRCA2 protein than the WT cells (Supplementary Fig. 2a) discarding BRCA2 variants protein levels as a possible cause for the reduction of the PLA signal observed in these cells.

**Fig. 3 | RAD51 efficient recruitment to nascent DNA requires BRCA2. a** (Top) Scheme of the assay and representative images of in situ PLA on nascent DNA between biotinylated EdU detected with anti-biotin antibody and RAD51-specific antibody DLD1 BRCA2-deficient cells (BRCA2[−/−]) or BRCA2[−/−] stably expressing either BRCA2 WT or the variants C315S (A7), S273L (C5), R3052W, and S3291A mutant, as indicated. Cells left untreated (UNT) or treated with HU (30′ or 1 h 0.2 mM) are shown. An individual signal is observed (focus) if the two probed proteins (RAD51 and EdU-Biotin) are in close proximity (<40 nm). For all the experiments we carried out two single-antibody control (anti-RAD51 and anti-biotin) to assess the specificity of the PLA signal. The scale bar indicates 10 μm. (Bottom) Quantification of RAD51 recruitment was measured as the number of PLA foci observed per nucleus. The data represent the mean + SEM with 200–300 cells analyzed in each experimental data set at each time point. The number of independent experiments performed was as follows: BRCA2 WT: (UNT $n = 9$, 5′ $n = 4$, 30′ $n = 8$, 1 h $n = 8$); C315S: (UNT $n = 7$, 5′ $n = 2$, 30′ $n = 4$, 1 h $n = 5$); S273L: (UNT $n = 3$, 30′ $n = 2$, 1 h $n = 2$); S3291A: (UNT $n = 4$, 5′ $n = 3$, 30′ $n = 2$, 1 h $n = 2$); R3052W: (UNT $n = 5$, 5′ $n = 3$; 30′ $n = 3$; 1 h $n = 3$); BRCA2[−/−]: (UNT $n = 3$, 5′ $n = 3$, 30′ $n = 3$, 1 h $n = 3$). Statistical difference was determined by the Kruskal–Wallis test followed by Dunn's multiple comparison tests; the $p$-values show significant differences compared to the BRCA2 WT clone. ns not significant, $^{**}p < 0.01$ ($p = 0.0058$ at NT BRCA2 WT vs. S3291A), $^{****}p < 0.0001$. **b** (Left) Representative images of in situ PLA on nascent DNA between biotinylated EdU detected with anti-biotin antibody and RAD51-specific antibody in DLD1 BRCA2 WT cells after 4 h Thymidine chase in cells left untreated or treated with HU (30′ 0.2 mM). (Right) Quantification of RAD51 recruitment measured as the number of PLA foci observed per nucleus after 4 h Thymidine chase in BRCA2 WT cells. The data represent the mean + SEM of two independent experiments with 200–300 cells analyzed in each experimental data set at each time point. Schemes of the PLA assay were created with BioRender.com. Source data are provided as a Source Data.

These results suggest that the NTD and the RAD51 binding site at the C-terminal region contribute to the recruitment of BRCA2 to nascent DNA.

## RAD51 efficient recruitment to nascent DNA requires BRCA2

BRCA2 is a loader of RAD51 at DSBs[3,22,44]. At stalled replication forks, BRCA2 protects the DNA from nucleolytic degradation, a function that is thought to be achieved by stabilizing RAD51 filaments through its C-terminal oligomeric-RAD51 binding site[6,7]; whether and how RAD51 loading by BRCA2 takes place at stalled forks intermediates remains poorly defined. *Xenopus* Brca2 has been previously reported as being required for Rad51 recruitment at replicative chromatin[12]. Because the mutated forms of BRCA2 affecting the different regions were less abundant than BRCA2 WT at active or stalled replication forks, we tested whether the recruitment of RAD51 was also altered in these cells. We monitored the RAD51-EdU PLA signal at different time points using the same conditions as for BRCA2-EdU PLA experiments (HU 0.2 mM). BRCA2-deficient cells showed 2-fold reduced recruitment of RAD51 to nascent DNA compared to BRCA2 WT cells (Fig. 3a). RAD51 recruitment to nascent DNA was delayed for cell lines bearing R3052W, S3291A, C315S, and S273L variants. RAD51 levels at nascent DNA also decreased in the thymidine chase experiment although to a lesser extent than BRCA2 indicating that RAD51 is bound to the chromatin as well as at the nascent DNA as previously shown[8,41] (Fig. 3b).

Together, these results suggest that all three domains are important for the localization of RAD51 at active and HU-stalled replication forks. BRCA2-deficient cells did not completely abrogate the RAD51-EdU PLA signal suggesting there is some BRCA2-independent recruitment of RAD51 to the nascent DNA.

## BRCA2 NTD, CTD, and the C-terminal RAD51 binding sites contribute to replication fork protection

BRCA2 stabilizes RAD51 filaments at stalled forks protecting them from nucleolytic degradation[7,8], defective protection results in stretches of ssDNA. Given the reduced recruitment to nascent DNA observed in cells expressing our variants, we next tested whether they were required for the protection of stalled forks using a DNA fiber assay. After two subsequent pulses of dNTP analogs IdU and CldU for 30 min to label nascent DNA as cells replicate, cells were treated with 5 mM HU for 4 h before fixation, a standard condition to reveal fork degradation/resection[7,45]. Under these conditions, DLD1 BRCA2-deficient cells showed decreased CldU/IdU ratio compared to BRCA2 WT cells suggesting fork degradation as expected, however, the reduction in this cell system was much less pronounced than in other cell systems reported. Cells bearing R3052W, the CTD variant that impairs HR, S3291A, and C315S also reduced the CldU/IdU ratio although to a lesser extent than the BRCA2-deficient cells so that the differences with BRCA2 WT were not significant (Fig. 4a). Fork degradation may induce fork restart defects. To investigate this, we modified the labeling set up

to monitor fork restart under the same treatment conditions. We performed the first labeling with IdU for 20 min followed by HU treatment and then released the cells into CldU for 20 min and monitored fork restart using DNA combing. As the DNA fiber assay, this method allows the analysis of single DNA molecules aligned on a slide; however, in this case, the DNA stretching is performed at a constant speed[46]. We found that ~50% of forks were able to restart upon release from HU in BRCA2 WT cells similar to previous reports[7,47]. This was also the case for BRCA2-deficient cells and cells bearing the BRCA2 C315S variant (Fig. 4b).

Together, these results suggest that all three domains, NTD, CTD, and the C-terminal RAD51 binding site protect stalled replication forks from aberrant nucleolytic degradation. Under these conditions, neither BRCA2-deficient cells nor BRCA2 C315S cells displayed defects in replication fork restart.

## BRCA2 contributes to the arrest of DNA replication under replication stress conditions

BRCA2-deficient cells challenged with mild doses of HU or multiple rounds of cisplatin fail to stall replication, which could be at the origin of ssDNA gap accumulation observed previously[13,14]. Given the sensitivity to HU of cells expressing BRCA2-C315S, we monitored the replication track length in HU conditions using DNA combing. Following the first pulse with IdU, we added CldU in the presence or absence of 0.5 mM HU for 2 h, as previously described[14]. Under these conditions, replication forks were stalled in BRCA2 WT cells resulting in a 5-fold reduction of CldU track length compared to the non-treated conditions (median: 32 μm in UNT vs. 6 μm in HU) (Fig. 5a.i). DLD1 BRCA2-deficient cells showed already a reduced track length in the absence of HU compared to BRCA2 WT cells indicating an overall slower replication in these cells as previously reported[48]. BRCA2-deficient cells also reduced the track length upon HU treatment although to a lesser extent (~2.5-fold) than BRCA2 WT cells (Fig. 5a.i) (median track length = 28 μm in UNT vs. 11 μm in HU). This defect was further revealed when representing the difference of the mean CldU track length between untreated and HU-treated conditions in the BRCA2-deficient cells compared to that in BRCA2 WT cells (26 in WT vs. 17 in BRCA2[−/−])(Fig. 5a.ii). All cells expressing BRCA2 C315S, R3052W, or S3291A also arrested the progression of the fork following HU treatment although there was a very mild defect in BRCA2-C315S bearing cells (3.4-fold reduction in track length compared to 5-fold reduction in BRCA2 WT cells). Interestingly, S273L cells showed 3.1-fold reduction in CldU track length, close to the levels of BRCA2-deficient cells suggesting a defective arrest.

These results suggest that in our isogenic cellular settings (DLD1 p53-mutated cell line), BRCA2 is partly required to arrest replication forks upon HU-induced replication stress. Moreover, neither the CTD nor the dsDNA binding activity of the NTD nor the C-terminal RAD51 binding site of BRCA2 seem specifically

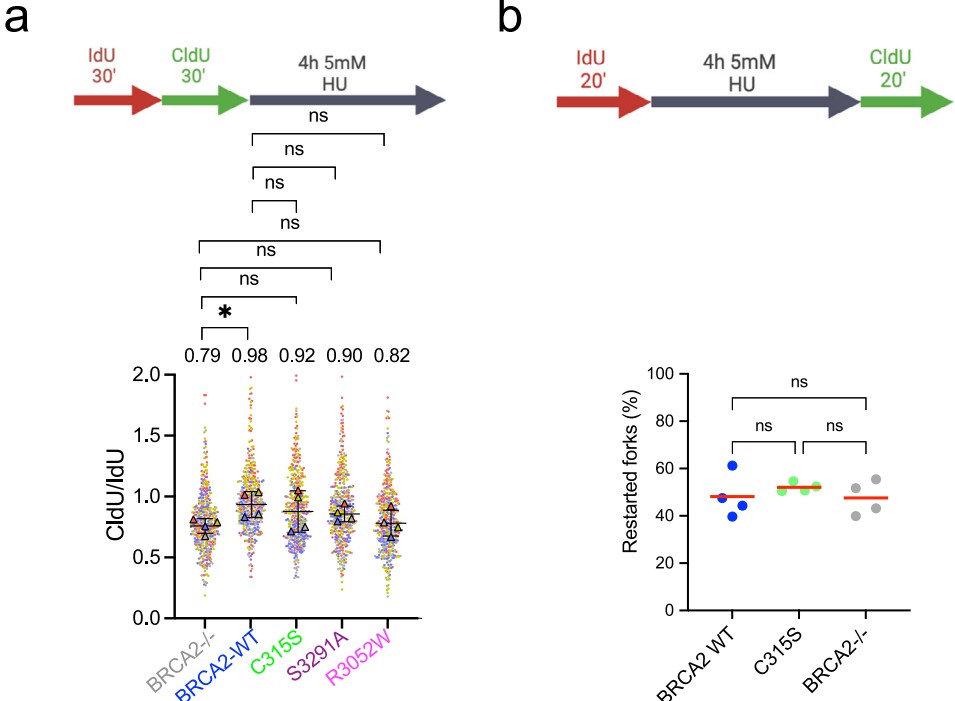

**Fig. 4 | Different domains of BRCA2 contribute to replication fork protection.**
**a** Labeling scheme of thymidine analogs (IdU and CldU) followed by HU treatment. DLD1-BRCA2-deficient cells complemented with the BRCA2 WT and mutated forms of BRCA2 were labeled with IdU (30 min) and then with CldU (30 min) followed by 4 h treatment with 5 mM HU, as indicated, after which cells were processed for DNA fiber analysis. Quantification of the track length ratio of CldU vs. IdU. Individual experiments are represented as a scatter plot with a different color from 100 replication tracks per data set. The super plot representation superimposes the summary statistics from the four independent experiments on top of the data from all cells. Differences between experiments were calculated using one-way ANOVA with Dunnett's multiple comparison tests on the mean of each experiment. ns, not significant, $*p < 0.05$ ($p = 0.0414$). **b** Schematic of the single-molecule DNA fiber tract analysis to detect fork restart after HU treatment. Quantification of restarting forks in BRCA2-deficient DLD1 cells, BRCA2 WT cells, or cells bearing the BRCA2 C315S variant, as indicated. Data are from four independent experiments; the percentage was calculated expressing the presence of CldU adjacent to IdU and was established on 200–500 tracks scored for each data set. The horizontal red line represents the mean. Statistics: Kruskal–Wallis test followed by Dunn's multiple comparison test. (ns, not significant). Labeling schemes created with BioRender.com. Source data are provided as a Source Data file.

## BRCA2 dsDNA binding activity is required to limit HU-induced ssDNA gaps but not PARPi-induced ssDNA gaps

BRCA2-deficient cells show high levels of ssDNA gaps that are accentuated under mild replication stress conditions such as treatment with 0.5 mM HU[12,14,16,20]. Given the HU sensitivity of cells expressing BRCA2-C315S and BRCA2-S273L, we wondered whether these cells accumulated ssDNA gaps under replicative stress. Hence, we subjected the cells to HU using previous conditions (Fig. 5a) now incorporating an extra step in the labeling scheme where we incubated the cells with S1 nuclease for 30 min (Fig. 5b top); this enzyme creates nicks in ssDNA regions without altering the dsDNA[47]. As described in previous reports in different cell systems[14,16,20], DNA tracks in DLD1 BRCA2-deficient cells displayed high sensitivity to S1 nuclease as manifested by the shortening of the CldU track length, this was in contrast to the BRCA2 WT cells where the track length was only mildly reduced (Fig. 5b). Importantly, CldU-labeled nascent DNA tracks from cells bearing the BRCA2-C315S variant showed high sensitivity to S1 treatment suggesting that these cells accumulate ssDNA gaps after mild HU-induced replication stress. This was also the case for the other NTD variant S273L. In stark contrast, the CldU track length in cells bearing BRCA2 R3052W (CTD mutant) or BRCA2 S3291A (RAD51 C-terminal binding site) did not vary compared to the untreated conditions. We also performed this experiment in unchallenged conditions but in this case, only BRCA2-deficient cells displayed detectable ssDNA gaps (Supplementary Fig. 5). These

results suggest that the NTD and specifically its dsDNA binding activity, impaired in BRCA2 C315S, is required to prevent ssDNA gap formation upon mild HU-induced replication stress.

Given that PARPi has been shown to generate ssDNA gaps[11] and the fact that only cells bearing R3052W but not C315S are sensitive to PARPi, we then assessed the presence of ssDNA gaps upon PARPi treatment (2 h 10 μM Olaparib) in these cells. As recently reported in a different cell system, BRCA2-deficient DLD1 cells showed high sensitivity to S1 treatment suggesting the accumulation of ssDNA gaps under these conditions. Similarly, cells bearing BRCA2-R3052W displayed ssDNA gaps. In stark contrast, the CldU track length of cells expressing BRCA2-C315S did not change, similarly to BRCA2 WT cells (Fig. 5c). These results perfectly correlate with the sensitivity of these cells to PARPi (Fig. 1c).

Together, cells expressing the NTD variants BRCA2-C315S and BRCA2-S273L display ssDNA gaps upon nucleotide depletion manifested by sensitivity to S1 nuclease that is not observed in cells bearing the R3052W CTD variant nor in cells expressing the C-terminus RAD51 binding mutant S3291A. In contrast, cells expressing R3052W CTD variant but not the NTD variants display PARPi-induced ssDNA gaps.

## BRCA2-C315S activates ATR/CHK1 upon RS but is deficient in ssDNA gap repair via HR

ssDNA gaps in BRCA2-deficient cells challenged with replication stress may arise due to a defect in arresting fork progression[14,49]. Fork arrest following a replication insult triggers the activation of the checkpoint kinase ATR/CHK1[50,51]. We monitored the checkpoint activation in our cells under the replication stress conditions used to detect ssDNA gaps

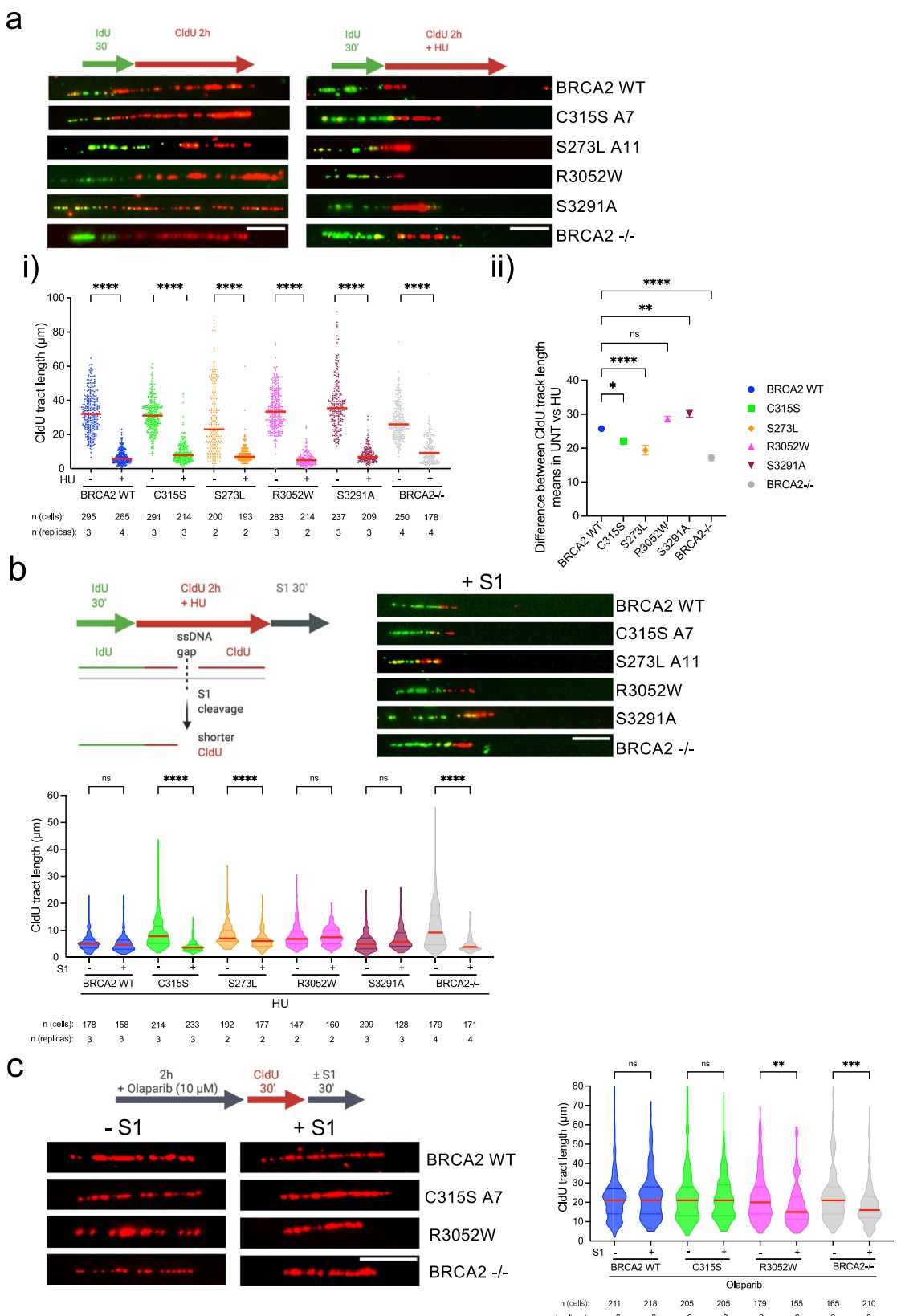

(0.5 mM HU for 2 h). In our cell system (DLD1 cells, p53 mutated), BRCA2-deficient cells displayed only a small reduction in the activation of ATR/CHK1 compared to BRCA2 WT cells as detected by the levels in pS345-CHK1 (Fig. 6a). BRCA2-C315S showed increased levels of pCHK1 compared to BRCA2-WT cells, consistent with the increased number of ssDNA gaps in these cells (Fig. 5b). This effect was not due to a net

increase in S-phase cells (Fig. 6b, Supplementary Fig. 6a); these results agree with the functional fork arrest observed in BRCA2-C315S cells upon HU treatment (Fig. 5a).

ssDNA gaps may also persist as a consequence of a defect in their repair. We have previously shown that the dsDNA binding activity of the NTD is specifically required to stimulate the recombination activity

**Fig. 5 | BRCA2 variants affecting dsDNA binding display ssDNA gaps despite being able to arrest replication upon HU treatment. a** (Top) Labeling scheme of thymidine analogs (IdU and CldU) in the absence or presence of HU and representative images of the replication tracks labeled as indicated from DLD1 BRCA2-deficient cells (BRCA2$^{-/-}$) or BRCA2$^{-/-}$ stably expressing either BRCA2 WT or the variants BRCA2-C315S (A7), BRCA2-S273L (A11), BRCA2-R3052W and BRCA2-S3291A mutant, as indicated, in unperturbed (left) or 0.5 mM HU-treated condition (right). The scale bar indicates 10 μm. i Quantification of CldU track length in the cell lines in (**a**). Data represent the median of two or more independent experiments per condition (details in the figure). Statistical difference was determined by the Kruskal–Wallis test followed by Dunn's multiple comparison test (all ****$p < 0.0001$), (the $p$-values show the significant differences compared to the untreated conditions). ii Quantification of the difference of the mean CldU track length in −HU vs. +HU. One-way ANOVA with Tukey's multiple comparison tests was performed on the difference between the means + SEM from an unpaired $t$-test calculated for each cell line separately. ns, not significant, *$p < 0.05$ ($p = 0.0310$), **$p < 0.01$ ($p = 0.0064$), all ****$p < 000.1$. **b** (Top left) Labeling scheme of thymidine analogs (IdU and CldU) in presence of HU followed by S1 nuclease treatment and schematic of the reduced CldU track length resulting from S1 cleavage at an ssDNA

region. (Top right) Representative images of the replication tracks labeled of the indicated cell lines in (a) in 0.5 mM HU treated condition followed by 30 min of S1 nuclease (or S1 buffer only) treatment, as indicated. The scale bar indicates 10 μm. (Bottom) Quantification of CldU track length in cells from (**b** top right). Data represent the median + 25% and 75% quartiles of two or more independent experiments per condition (details in the figure). Statistical difference was determined by the Kruskal–Wallis test followed by Dunn's multiple comparison test (ns, not significant, all ****$p < 0.0001$). See also Supplementary Fig. 5. **c** (Left) Labeling scheme and representative images of the replication tracks labeled as indicated from BRCA2-deficient cells (BRCA2$^{-/-}$) alone or stably expressing either BRCA2-WT, BRCA2-C315S (A7), BRCA2-R3052W, in 10 μM Olaparib treated condition followed by 30 min of S1 nuclease (or S1 buffer only) treatment, as indicated. The scale bar indicates 10 μm. (Right) Quantification of CldU track length in cells from (**c** left). Data represent the median + 25% and 75% quartiles of two independent experiments with the number of fibers analyzed detailed in the figure: Statistical difference was determined by the Kruskal–Wallis test followed by Dunn's multiple comparison test. ns, not significant, **$p < 0.01$ ($p = 0.0035$), ***$p < 0.001$ ($p = 0.0007$). Labeling schemes were created with BioRender.com. Source data are provided as a Source Data file.

of RAD51 in vitro at dsDNA/ssDNA containing DNA substrates and that this activity is defective in the NTD fragment mutated at C315S in vitro (BRCA2$_{NTD-C315S}$). Using an ssDNA substrate, BRCA2$_{NTD-C315S}$ showed intact RAD51-mediated DNA strand exchange stimulation. These results suggested that BRCA2 dsDNA binding activity was required to stimulate recombination at dsDNA-containing substrates such as resected DNA or ssDNA gaps. We also showed that BRCA2$_{NTD}$ could bind gapped DNA substrates in vitro[28]. Given that the repair of ssDNA gaps by template switching is thought to involve BRCA2 and RAD51[18,19,24,25,52] we wondered whether BRCA2-C315S cells were defective in the repair of ssDNA gaps. To test this hypothesis, we produced and purified BRCA2$_{NTD}$ and BRCA2$_{NTD-C315S}$ fragments from human cells, RAD51, and RPA from bacteria and used synthetic radio-labeled oligonucleotides that mimic an ssDNA gap to reconstitute an ssDNA gap recombination repair reaction in vitro. For comparison, we performed a 3′-tail reaction using the same donor dsDNA sequence to avoid sequence-dependent effects. To generate the ssDNA gap substrate, we used a set of three synthetic oligonucleotides that anneal at the two ends of a 167mer leaving an ssDNA stretch of 83 nucleotides (nt) (gap) in the middle. In this reaction, the ssDNA gap or the tailed substrate is first coated with RPA, and RAD51 is subsequently incubated with this complex in the presence or absence of BRCA2$_{NTD}$ (or BRCA2$_{NTD-C315S}$) before adding the radiolabeled dsDNA donor. As expected, in the absence of RPA, RAD51 could perform DNA strand exchange on this synthetic-tailed substrate whereas when RPA was allowed to bind first, the reaction was strongly inhibited (Supplementary Fig. 6b). The same was true for the gapped DNA (Fig. 6c). As there is no 3′-overhang in this substrate and the dsDNA donor contains blunt ends, this result indicates that RAD51 can readily invade the template strand from an ssDNA gap without the need of an ssDNA 3′-end.

As previously shown for the tailed-substrate with a different dsDNA donor[28], BRCA2$_{NTD}$ stimulated RAD51-driven strand exchange reaction overcoming RPA inhibition whereas BRCA2$_{NTD-C315S}$ did not (Supplementary Fig. 6b). Importantly, BRCA2$_{NTD}$ was also able to stimulate RAD51-mediated recombination at ssDNA gap mimicking substrates by 2-fold in a concentration-dependent manner whereas BRCA2$_{NTD-C315S}$ could not stimulate the reaction even at the highest concentration tested (Fig. 6c). We previously showed that the CTD alone stimulates poorly RAD51-mediated DNA strand exchange in these experimental conditions[28] and therefore was not tested.

In conclusion, BRCA2-C315S expressing cells display functional ATR/CHK1 checkpoint activation. BRCA2$_{NTD}$ stimulates RAD51-mediated ssDNA gap repair in vitro, a function that is impaired in BRCA2$_{NTD-C315S}$.

## BRCA2-C315S cells display increased chromatid gaps in metaphase

ssDNA gaps near arrested replication forks may persist through mitosis or can be converted into DSBs[15,53]. These lesions are expected to cause structural chromosomal aberrations such as chromatid breaks or complex chromosomal aberrations like radials and chromosome fusions which are well documented in HU-treated BRCA2-deficient cells[6,48,54]. We thus analyzed metaphase chromosome spreads in our BRCA2-C315S-mutated cell lines either left untreated or treated with mild HU (0.5 mM 2 h) or acute HU (5 mM for 5 h) before releasing them into colcemid. As expected, DLD1 BRCA2-deficient cells displayed an increased number of chromosomal aberrations already in untreated conditions[27]. This phenotype was mainly contributed by chromosome gaps, radial chromosomes, and fusions and it was exacerbated in the HU-treated cells (Fig. 7ai). In contrast, the levels of chromosomal aberrations in DLD1 BRCA2 WT cells were very limited (about 1 aberration per metaphase) in untreated conditions and the average number of aberrations did not change with HU although the counts per metaphase increased. Interestingly, cells bearing BRCA2-C315S showed an increased number of chromosomal aberrations in unchallenged conditions compared to BRCA2 WT cells and the difference was further accentuated in HU-treated cells (Fig. 7a i). Remarkably, the number of chromatid gaps per metaphase spread was as abundant in BRCA2-C315S cells as in BRCA2-deficient cells in unperturbed conditions or after treatment with mild replication stress (0.5 mM HU 2 h) or the acute dose of HU (5 mM 5 h), consistent with the presence of replication-associated gaps in these cells (Fig. 7a ii, Fig. 5b).

Finally, replication-associated gaps may result in regions of unreplicated DNA that could lead to chromatin bridges in anaphase. Therefore, we analyzed the presence of chromosome segregation errors by looking at anaphase bridges as previously shown in BRCA2-deficient cells[54,55]. We found a slight increase in the number of cells with anaphase bridges in the BRCA2-C315S cell line compared to BRCA2 WT cells measured in unperturbed conditions which were accentuated in BRCA2-deficient cells (Fig. 7b). However, the levels of anaphase bridges did not increase upon mild HU treatment neither in the BRCA2-deficient cells nor in BRCA2-C315S cells suggesting that these structures may not be a direct consequence of ssDNA gaps (Supplementary Fig. 7).

Together, these results suggest that cells expressing the BRCA2-C315S variant accumulate replication-associated ssDNA gaps that persist through mitosis as manifested in metaphase spreads. The levels of ssDNA gaps that accumulate in cells expressing BRCA2-C315S in mitosis are comparable to those found in BRCA2-deficient cells.

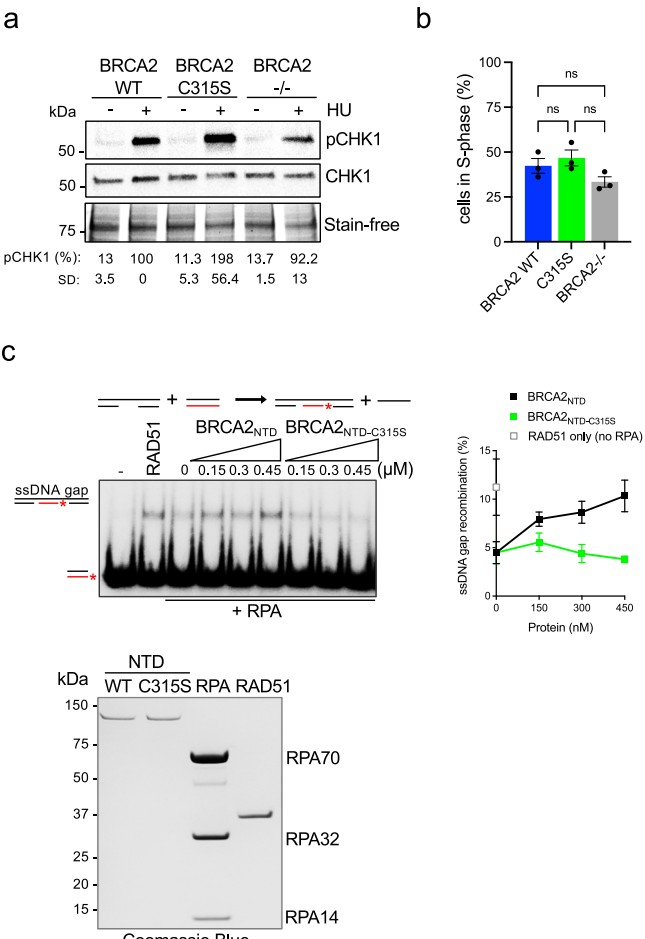

**Fig. 6 | BRCA2-C315S cells show an active ATR checkpoint whereas NTD-C315S fragment shows reduced stimulation of RAD51-mediated ssDNA gap repair.**
**a** (Left) Western blot showing the phosphorylation of CHK1 (pCHK1) after exposure to HU (0.5 mM, 2 h) as indicated in DLD1 BRCA2-deficient cells (BRCA2$^{-/-}$) or BRCA2$^{-/-}$ cells stably expressing BRCA2 WT or the variant BRCA2-C315S. Stain-free cropped gel is used as the loading control. pCHK1 levels relative to the total CHK1 signal are shown below the blots, results are presented as a percentage of pCHK1/CHK1 compared to the BRCA2 WT clone treated with HU. The data represent the mean ± SD of two independent experiments. **b** Frequency of S-phase cells in DLD1 BRCA2-deficient cells (BRCA2$^{-/-}$) or BRCA2$^{-/-}$ cells stably expressing BRCA2 WT or the variant BRCA2-C315S. The data represent the mean ± SEM of three independent experiments. Statistical difference was determined by the Kruskal–Wallis test followed by Dunn's multiple comparison test (ns, not significant). See also Supplementary Fig. 6. **c** (Left) DNA strand exchange reaction using an ssDNA gap mimicking substrate (Table S1) in the presence or absence of RPA, RAD51, and increasing concentrations of BRCA2$_{NTD-WT}$ or BRCA2$_{NTD-C315S}$, as indicated. (Right) Quantification of the reaction on the left. Data represent the mean from three independent experiments. Error bars SD. (Bottom) SDS–PAGE showing purified BRCA2$_{NTD-WT}$ (1 μg), BRCA2$_{NTD-C315S}$ (1 μg), RPA (3 μg), and RAD51 (1.5 μg) used in the DNA strand exchange reactions. Source data are provided as a Source Data file.

## Discussion

Here we report that cells bearing a single amino acid variant of BRCA2, S273L, or C315S, that impair the DNA binding activity of BRCA2 at its N-terminal DNA binding domain (NTD)(Fig. 1b)[28], are highly sensitive to replication stress induced by HU. On the other hand, a pathogenic variant affecting the canonical DNA binding domain (CTD), R3052W[30], was resistant to HU treatment. Unlike R3052W, defective in DSB repair by HR and hypersensitive to PARPi[30], variants at the NTD were resistant to PARPi and HR proficient based on a gene-targeting reporter assay[37] (Table 1).

Consistent with the sensitivity to HU, cells expressing the C315S and S273L variants reduced the localization of BRCA2 and RAD51 at both unperturbed or HU-challenged replication forks; this phenotype was also observed although to a different degree in the other variants/mutants analyzed. Cells expressing R3052W were highly sensitive to ICLs inducing agent MMC and resistant to HU as recently observed with another compound mutation at the CTD[56]. However, unlike the compound mutation in the previous report, R3052W impaired HR. The strong effect of this pathogenic mutation might be due to the combined impact of R3052W on DNA binding and its predicted destabilization effect on the interface between oligonucleotide binding folds (OB) 2 and OB3 of the CTD[29,57]. Indeed, cells bearing this variant also display a growth defect compared to the other cell lines used in this study.

DLD1 BRCA2-deficient cells did not fully arrest replication forks upon mild replication stress (0.5 mM HU, 2 h) when assessed by DNA combing, although the effect was much more modest than the one reported[14]. This discrepancy may arise from the different cell systems utilized (DLD1 cells are p53 mutated and deficient in mismatch repair). However, it is important to note that the comparison between HU-treated cell lines might give rise to false interpretations as the basal replication fork track length among different cell lines might differ as we have observed. We, therefore, compared the difference of the mean between untreated and HU-treated conditions for each clone to compare the cell lines (Fig. 5a.ii). In doing so, we found that only one of the cell lines, the one expressing S273L, displayed a significant reduction in fork arrest capacity close to one observed in the BRCA2-deficient cells. These results suggest that BRCA2 contributes partially to fork arrest upon HU treatment and that the ssDNA binding activity of the NTD, impaired in cells bearing S273L, might be involved in this function.

Interestingly, despite the normal replication arrest and concomitant ATR activation, cells expressing BRCA2-C315S accumulated ssDNA gaps following replication stress as manifested by the sensitivity to S1 nuclease treatment and observed in metaphase spreads. The fact that BRCA2-C315S cells did not show a significant number of ssDNA gaps in unperturbed conditions whereas they were detected in similar numbers to those of the BRCA2 deficient cells in metaphase spreads suggests that the DNA combing and S1 nuclease may only detect a fraction of ssDNA gaps present in cells.

Overall, these findings are consistent with the recently proposed role of BRCA2 in gap suppression[14] whereas it challenges the idea that fork arrest alone underlays gap suppression. Given that the NTD variant C315S is defective in dsDNA binding while preserving its ssDNA binding activity, these results strongly suggest that the dsDNA binding activity of BRCA2, unique to the NTD[28], is specifically required to suppress replication ssDNA gaps. Cells expressing S273L altering both ssDNA and dsDNA binding also presented ssDNA gaps reinforcing the idea that the NTD is involved in gap suppression. Moreover, the fact that all three variants tested showed similar defects in fork protection but neither cells expressing S3291A nor R3052W exhibited ssDNA gaps upon nucleotide depletion in this isogenic setting suggest that the fork protection function of BRCA2 can be uncoupled from its ssDNA gap suppression activity as recently proposed[11]. These results are also consistent with a recent report in which the mutant S3291A in Chinese hamster cells was found devoid of ssDNA gaps[14].

ssDNA gaps in BRCA2-deficient cells have been proposed to arise from PrimPol repriming[13,21] or Okazaki fragment processing defects[11,58]. The fact that we could not observe a substantial defect in replication fork arrest nor fork restart in cells bearing C315S is consistent with the latter however it requires future investigation.

ssDNA gaps are a substrate for repair by homologous recombination[19,25,52]. Reconstitution of the ssDNA gap repair reaction in vitro indicated that RAD51 can perform strand invasion in this context without the need for a resected 3'-overhang, consistent with

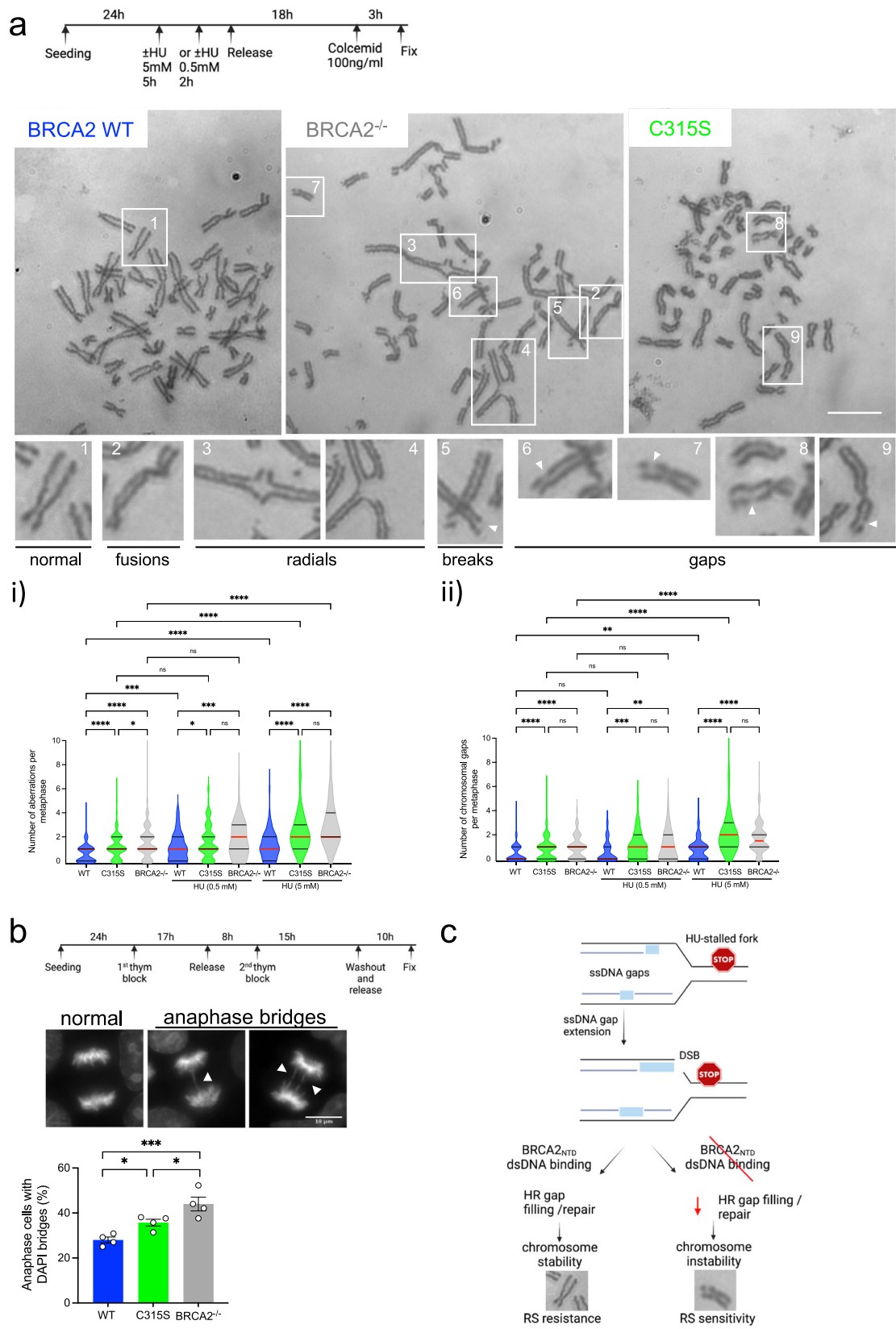

the template switch model for repair[18]. Moreover, we found that in the presence of RPA-coated ssDNA, the NTD of BRCA2 could stimulate RAD51 recombination activity at ssDNA gaps. Interestingly, the capacity of BRCA2$_{NTD-C315S}$ to promote recombination in ssDNA gaps-mimicking substrates in vitro was strongly reduced suggesting that the dsDNA binding activity of BRCA2 promotes ssDNA gap repair.

Interestingly, despite the HU-induced accumulation of ssDNA gaps, cells bearing the BRCA2-C315S variant showed resistance to PARPi, a chemotherapeutic drug that has been reported to accelerate replication[59] and cause ssDNA gaps in BRCA1/2-deficient cells[11]. Importantly, analysis of the presence of ssDNA gaps in cells treated with PARPi indicated that cells expressing C315S do not accumulate

**Fig. 7 | BRCA2-C315S cells accumulate gaps in metaphase chromosomes. a** (Top) Schematic representation of the experiment timing and the dose of HU used to detect chromosomal aberrations. (Bottom) Representative images of metaphase spreads of DLD1 BRCA2-deficient cells (BRCA2$^{-/-}$) or BRCA2$^{-/-}$ cells stably expressing BRCA2 WT or BRCA2-C315S (A7), as indicated, treated with 5 mM HU for 5 h. The type of chromosomal aberrations observed is indicated with numbers and magnified below. The scale bar indicated 10 μm. i Quantification of global chromosomal aberrations from the same cells either left untreated or upon treatment with HU (0.5 mM for 2 h or 5 mM for 5 h), as indicated. Statistical difference was determined by the Kruskal–Wallis test followed by Dunn's multiple comparison test. ns, not significant, *$p < 0.05$ ($p = 0.0129$ in C315S vs. BRCA2$^{-/-}$, and $p = 0.044$ in WT 0.5 mM HU vs. C315S 0.5 mM HU), ***$p < 0.001$ ($p = 0.0002$ in WT vs. WT 0.5 mM HU, and $p = 0.0003$ in WT 0.5 mM HU vs. BRCA2$^{-/-}$ 0.5 mM HU, ****$p < 000.1$). ii Quantification of chromosomal gaps observed in the same cell lines. Data in (i) and (ii) represent the median and 25% and 75% quartiles of three independent experiments where 39–50 metaphase spreads were analyzed in each experimental data set. Only metaphases with at least 30 chromosomes were considered. Statistical difference was determined by the Kruskal–Wallis test followed by Dunn's multiple comparison test. ns not significant, **$p < 0.01$ ($p = 0.0011$ in WT 0.5 mM HU vs. BRCA2$^{-/-}$ 0.5 mM HU, and $p = 0.0019$ IN WT vs. WT 5 mM HU), ***$p < 0.001$ ($p = 0.0002$), ****$p < 000.1$). **b** (Top) Synchronization scheme and representative images of normal chromosome segregation stained with DAPI and type of aberrant chromosome segregation (DAPI bridges) that were observed in the DLD1 BRCA2 deficient cells (BRCA2$-/-$) or BRCA2$-/-$ cells stably expressing BRCA2 WT or BRCA2-C315S (A7). The scale bar indicates 10 μm. (Bottom) Quantification of cells with aberrant chromosome segregation in BRCA2$^{-/-}$ cells and in the BRCA2$^{-/-}$ clones stably expressing BRCA2 WT, BRCA2-C315S, as indicated. Data represent the mean and SEM of four independent experiments: at least 150 anaphase cells were analyzed in each experimental data set. A two-way ANOVA test with Tukey's multiple comparisons test was used to calculate the statistical significance of differences (normal vs cells with anaphase bridges, only cells with anaphase bridges are plotted in the graph). *$p < 0.05$ ($p = 0.048$ in WT vs. C315S, and $p = 0.0325$ in C315S vs. $-/-$), ***$p < 0.001$ ($p = 0.0001$). See also Supplementary Fig. 7. **c** Working model for the role of BRCA2 in the repair of replication-associated gaps following RS. Light blue rectangles represent ssDNA gaps. See text for details. Synchronization scheme and the model were created with BioRender.com. Source data are provided as a Source Data file.

PARPi-induced ssDNA gaps consistent with their resistance to PARPi. In contrast, cells expressing R3052W that were sensitive to PARPi displayed ssDNA gaps under these conditions. These findings reinforce the idea that PARPi-induced ssDNA gaps correlate with PARPi sensitivity as previously reported[11]. Similarly, in our conditions, DSB repair capacity also correlates with PARPi resistance although in other works these two processes have been uncoupled. Along these lines, we show that HU-related gap suppression is separable from DSB repair capacity as cells expressing BRCA2-C315S or S273L are proficient in HR-mediated DSB repair but sensitive to HU. Moreover, ssDNA gaps do not necessarily result in DSBs as we mainly observe chromatid discontinuities without misalignment in metaphase spreads indicative of ssDNA gaps[60]. Interestingly, as two different variants led to gaps in the presence of one agent but not the other, these results suggest that nucleotide depletion and PARPi induce different types of ssDNA gaps that in turn require distinct gap-filling mechanisms. These differences might be due to the nature of the DNA ends and/or the incorporation of ribonucleotides as has been described for the former[61]. Based on our results we propose that HU-induced ssDNA gaps are filled-in preferentially via HR/TS mechanisms.

Putting together our in vitro and cell-based results and our previous findings[28] in the context of the literature we propose a model (Fig. 7c) in which (a) BRCA2 localizes at unperturbed and stalled replication forks participating in the loading of RAD51 at these sites. (b) Upon nucleotide depletion-induced replication stress, BRCA2 protects replication forks from nucleolytic degradation, a function that is probably achieved through different domains including both DNA binding domains, CTD and NTD, and the C-terminal RAD51 binding site. (c) In contrast, BRCA2 gap suppression activity is particularly dependent on its dsDNA binding activity, located at the NTD which promotes the repair of the resulting replication-associated ssDNA gaps by RAD51. (d) Unrepaired replication-associated lesions in BRCA2-C315S cells lead to abundant chromatid discontinuities or gaps, especially in acute replication stress conditions, explaining their HU sensitivity.

Based on our results with BRCA2-C315S, BRCA2-S273L, and BRCA2-R3052W cells, we propose that the repair of DSBs requires the ssDNA binding activity via the canonical CTD of BRCA2 whereas the NTD is dispensable/redundant for this function.

Our data are consistent with C315S and S273L being separation of function variants defective in ssDNA gap suppression and replication-associated ssDNA gap repair/fill-in but not in the repair of DSBs.

Our findings may have clinical implications for the assessment of variants of unknown clinical significance (VUS) located at the NTD as defects in HR-repair of replicative lesions would not be picked up by the current methods to assess HR proficiency[30,57,62] as exemplified in the case of BRCA2-C315S or BRCA2-S273L, but may nonetheless be linked to cancer predisposition given the genome instability observed in cells bearing these variants.

## Methods

### Plasmids
Human 2XMBP-BRCA2$_{250-500}$ and EGFP-MBP-BRCA2 subcloning in phCMV1 expression vector were generated as described[28,63].

### Site-directed mutagenesis
Point mutations (C315S, S273L, S3291A) were introduced in 2xMBP-BRCA2$_{250-500}$, EGFP-MBP-BRCA2 vector using QuikChange II and QuikChange XL site-directed mutagenesis kit (Agilent Technologies), respectively, as previously described[28,30].

Mutagenesis primers were designed using the QuickChange Primer Design program and purchased from MWG Eurofins. All mutations were verified by Sanger sequencing.

### Cell lines, cell culture
The human cell lines HEK293-T cells (gift from Dr. Mounira Amor-Gueret) were cultured in DMEM (Eurobio Abcys, Courtaboeuf, France) media containing 25 mM sodium bicarbonate and 2 mM L-glutamine supplemented with 10% FBS (EuroBio Abcys). The BRCA2-deficient colorectal adenocarcinoma cell line DLD1 BRCA2$^{[-/-31}$ (HD 105-007) and the parental cell line DLD1 BRCA2$^{+/+}$ (HD-PAR-008) were purchased from Horizon Discovery (Cambridge, England). The cells were cultured in RPMI media (EuroBio Abcys) containing 25 mM sodium bicarbonate and 2 mM L-glutamine (EuroBio Abcys) Supplemented with 10% FBS (EuroBio Abcys). The DLD1 BRCA2$^{-/-}$ cells were maintained in growth media containing 0.1 mg/ml hygromycin B (Thermo Fisher Scientific). The stable cell lines of DLD1$^{-/-}$ BRCA2-deficient cells expressing BRCA2 WT or variants of interest generated in this study were cultured in growth media containing 0.1 mg/ml hygromycin B (Thermo Fisher Scientific) and 1 mg/ml G418 (Sigma-Aldrich).

All cells were cultured at 37 °C with 5% $CO_2$ in a humidified incubator and all cell lines used in this study have been regularly tested for mycoplasma contamination (MycoAlert, Lonza) and genotyped using GenePrint kit (Promega).

### Stable cell line generation
To generate DLD1 BRCA2$^{-/-}$ stable cell lines expressing human BRCA2 variants of interest we transfected one 100 mm plate of DLD1 BRCA2$^{-/-}$ cells at 70% of confluence with 10 μg of a plasmid containing human EGFP-MBP-tagged BRCA2 cDNA (carrying mutation of interest) using

TurboFect (Thermo Fisher Scientific) according to manufacturer's instructions; 48 h post-transfection the cells were serial diluted and cultured in media containing 0.1 mg/ml hygromycin B (Thermo Fisher Scientific) and 1 mg/ml G418 (Sigma-Aldrich) for selection. Single-cell colonies were isolated and later expanded and their genomic DNA was extracted to verify the mutation by sequencing. BRCA2 protein levels were detected by Western Blot using BRCA2 antibody (1:1000, OP95, EMD Millipore).

## Western blotting

Cellular pellet was lysed in lysis buffer (50 mM HEPES pH 7.5, 250 mM NaCl, 5 mM EDTA, 1% NP-40,1 mM DTT, 1 mM PMSF, 1X protease inhibitor cocktail (Roche)) and cells were incubated on ice for 30 min, vortexed every 5 min. Lysates were centrifuged at $18,000 \times g$ for 1 h at 4 °C. The supernatant was transferred to a prechilled Eppendorf tube and stored at −80 °C. For protein electrophoresis, samples were heated in 1× SDS sample buffer for 5 min at 95 °C, loaded on a stain-free 4−15% SDS gel (Bio-Rad), and migrated at 130 V for 90 min in running buffer (1x Tris-Glycine, 0.1% SDS). The stain-free gel was visualized using a ChemiDoc camera (Bio-Rad). For transfer, a nitrocellulose membrane (VWR) was pre-equilibrated in dH$_2$O and transfer buffer (1x Tris-Glycine, 0.025% SDS, 10% methanol). The proteins were transferred for 2 h at 0.35 A at 4 °C. The membrane was blocked in 5% milk in 1× TBS-T at room temperature for 30 min and then incubated with the respective antibody (see antibodies below) in 5% milk in 1× TBS-T overnight at 4 °C. After extensive washes in TBS-T ($3 \times 10$ min), the membrane was incubated for 1 h with the appropriate secondary HRP-antibody at room temperature on a shaker. After 3 more washes in TBS-T, the membrane was developed using ECL prime western blotting detection reagent (VWR) and visualized using a ChemiDoc camera (Biorad).

## Antibodies used for western blotting

Mouse anti-MBP (1:5000, R29, Cat. #MA5-14122, Thermo Fisher Scientific), mouse anti-BRCA2 (1:1000, OP95, EMD Millipore), mouse anti-CHK1 (1:1000, Cat. #2360, Cell Signaling Technology), rabbit anti-pCHK1-S345 (1:500, Cat #2348, Cell Signaling Technology), Horseradish peroxidase (HRP) conjugated secondary antibodies used: mouse-IgGκ BP-HRP (IB: 1:5000, Cat. #sc-516102, Santa Cruz), HRP Goat anti-mouse IgG (1:10,000, Cat #115-035-003, Jackson Immuno), HRP Goat anti-rabbit IgG (1:10,000, Cat #111-035-003, Jackson Immuno).

## Protein purification

Wild-type and mutant human 2×MBP-BRCA2$_{NTD}$ fragment (BRCA2 aa 250−500) cDNAs were purified as described previously[28]. Briefly, $10 \times 15$-cm plates of HEK293 cells were transiently transfected using Turbo-Fect (Thermo Scientific) following the manufacturer's specifications and harvested 30 h post-transfection. Cell extracts were bound to amylose resin (NEB), and the protein was eluted with 10 mM maltose. The eluate was further purified by ion exchange using BioRex 70 resin (Bio-Rad) and step eluted at 250, 450, and 1 mM NaCl. Each fraction was tested for nuclease contamination. The CTD of BRCA2 and RAD51 were purified as described before[28] Only the nuclease-free fractions were used for EMSA or DNA strand exchange assays.

RPA was expressed from plasmid p11d-tRPA (kind gift from Marc Wold) in BL21(DE3) cells (Novagen) and purified as described[64].

## Cell survival and viability assays

Clonogenic survival assay was assessed in DLD1 BRCA2$^{+/+}$ expressing the endogenous BRCA2 protein, DLD1 BRCA2-deficient cells (BRCA2$^{-/-}$) or DLD1 BRCA2-deficient cells stably expressing either EGFP-MBP-BRCA2 WT or different clones expressing the variants (C315S, S273L, and R3052W). Cells seeded at 70% of confluence were treated either with MMC (Sigma-Aldrich) at concentrations: 0, 0.5, 1.0, and 2.5 µM or

with HU (Sigma-Aldrich) at concentrations 0, 1, 5, or 10 mM. After 1 h (MMC) or 24 h (HU) treatment, the cells were serially diluted in normal growth media/RPMI (Eurobio) and seeded at 100, 250, 500, 1000, or 10,000 cells in triplicates into six-well plates depending on the drug concentration. The media was changed every third day, after 10−12 days in culture the plates were stained with crystal violet (Sigma Aldrich) and colonies were counted. The surviving fraction was determined for each drug concentration as compared to the non-treated condition of the same clone.

## MTT assay

Cell viability was assessed in DLD1 BRCA2$^{+/+}$ expressing the endogenous BRCA2 protein, DLD1 BRCA2-deficient cells (BRCA2$^{-/-}$) or DLD1 BRCA2-deficient cells stably expressing either EGFP-MBP-BRCA2 WT or different clones expressing the variants (C315S, S273L, and R3052W). The cells were seeded at 2000−4000 cells per well depending on the clone and treated at increasing concentrations of Olaparib (AZD2281, Selleck Chemicals) 0.5, 1.0, and 2.5 µM for 6 days. On the 6th day, the media was removed and cells were washed with 1× PBS. Cell viability was assessed with 3-[4,5-dimethylthiazol-2-yl]-2,5-diphenyltetrazolium bromide (MTT, #M5655, Sigma Aldrich). The solution was removed and MTT crystals were dissolved in 100 µl 100% DMSO (Sigma-Aldrich). The absorbance was read in a microplate reader at 570 nm. The calculation was corrected for the absorbance of the blank (DMSO only) and the survival percentage was calculated by dividing the absorbance into the cells treated by the absorbance obtained in the untreated cells.

## Proximity ligation assay on nascent DNA

500,000 cells were seeded on glass coverslips the day before the experiment to reach 70% confluence. The next day, cells were pulse-labeled with 25 µM EdU (Thermo Fisher) for 10 min. In the case of HU treatment, cells were washed once with 1× PBS and incubated with 0.2 mM HU at 5 min, 30 min, and 1 h time points. For thymidine chase experiments, cells were washed with 1× PBS 3 times and incubated in a medium supplemented with 125 µM thymidine (Sigma-Aldrich) for 4 h. After treatment and labeling cells were washed 3 times with 1× PBS and put on ice for another wash with cold PBS. Cells were then incubated and washed once with CSK buffer (10 mM PIPES, pH 6.8, 0.1 M NaCl, 0.3 M sucrose, 3 mM MgCl$_2$, EDTA-free Protease Inhibitor Cocktail (Roche)) followed by CSK-T (10 mM PIPES, pH 6.8, 0.1 M NaCl, 0.3 M sucrose, 3 mM MgCl$_2$, EDTA-free Protease Inhibitor Cocktail (Roche), 0.5% Triton X-100) incubation for 5 min at RT and one CKS-T wash. The last two washes were done once with CSK and once with PBS followed by fixation with 4% paraformaldehyde (Euromedex) for 20 min at RT. Cells were once again washed with PBS and blocked for 1 h in PBS + 0.1% Tween + 5% BSA at RT. The coverslips containing the cells were incubated with 25 µl of the click reaction mix (PBS 1x, 6 nM biotin azide, 10 mM Sodium Ascorbate, and 2 mM CuSO$_4$) for 30 min at RT in a light-protected chamber. Cells were then washed twice in PBS + 0.1% Tween + 5% BSA followed by primary antibody incubation overnight at 4 °C (see antibodies below). The next day, the samples were subjected to the standard PLA protocol (Sigma-Aldrich Duolink) where: cells were first rinsed in 2 ml of 1X Wash Buffer A followed by $2 \times 10$ min incubation of coverslips in 2 ml of 1× wash buffer A on a shaker. PLA probes were prepared according to primary antibody species, vortexed, and incubated for 20 min at RT. 25 µl of probe mix was added to each coverslip and incubated for 1 h at 37 °C in a pre-heated humid chamber. For the positive control, PLA Mouse/Rabbit plus and minus probes were used for each Mouse/Rabbit antibody, separately. For negative control, only one primary antibody was used together with PLA Mouse plus and Rabbit minus probe. Cells were again rinsed in 2 ml of 1× Wash Buffer A followed by $2 \times 10$ min incubation of coverslips in 2 ml of 1× Wash Buffer A on a shaker before the ligation step. The ligation mix was prepared according to the manufacturer's specifications and the mix was added to the coverslip and incubated for

30 min at 37 °C in the pre-heated humid chamber. Washing was done as in previous steps with 1× wash buffer A 2 × 10 min, prior to the amplification reaction. The amplification mix was prepared added to the coverslips and incubated for 100 min at 37 °C in the pre-heated humid chamber. After the amplification step, coverslips were rinsed in 2 ml of 1× wash buffer B followed by 2 × 10 min incubation of coverslips in 2 ml of 1× Wash Buffer B on a shaker. The final wash was done using diluted 0.01× Wash Buffer B. Coverslips were let to air-dry for 5–10 min and were mounted with 7 μl ProLong Diamond (Invitrogen) onto coverslips with clear nail polish.

Images were acquired using a DM6000B upright widefield Microscope (Leica) equipped with an ×63 Plan Apochromat oil immersion objective (Leica, NA: 1.4). The fluorescence signal was recorded with bloc filters. TX2 emission was detected at 604–644 nm upon excitation between 542–582 nm. DAPI emission was detected at 445–495 nm upon excitation between 375 and 435 nm. Images were obtained with an sCMOS Orca Flash 4.0 camera (Hamamatsu). The whole system was driven by Metamorph (Molecular Devices). For 3D imaging, stacks were acquired with a z-step of 1 μm. The number of PLA spots was counted with a customized macro where the nucleus was defined by a minimum pixel size of 1500 on DAPI and a mask was generated and applied to the Z-projection to count the spots within the nucleus. The PLA spots were quantified using the ImageJ plugin Find Maxima in the Z-projection with a prominence of 2000.

Primary antibodies used for PLA were as follows: mouse anti-BRCA2 (1:500 EMD Millipore Cat. # OP95), rabbit anti-biotin (1:3000 Bethyl laboratories Cat. # BETA150-109A), mouse anti-biotin (1:3000 Jackson ImmunoResearch Cat. # AB_2339006), mouse anti-RAD51 (1:500 Novus Biologicals Cat. # NB100-148), mouse anti-PCNA (1:500 Santa Cruz Biotechnology Cat. # sc-56) and mouse anti-histone H1 (1:500 Santa Cruz Biotechnology Cat. # sc-8030).

### DNA fiber assay

DNA fiber labeling scheme to visualize replication fork degradation was performed as previously described[65]. Briefly, cells were labeled with 25 μM IdU, washed with warm media, exposed to 50 μM CldU, washed again with warm media, and treated with 5 mM hydroxyurea for 4 h. Cells were lysed and DNA fibers were stretched onto glass slides and then dried and fixed in methanol/acetic acid (3:1) for 10 min. The DNA fibers were denatured with 2.5 M HCl for 1 h, washed with PBS, and blocked with 2% BSA in PBS-Tween for 60 min. IdU replication tracts were revealed with a mouse anti-BrdU/IdU antibody from BD Biosciences (347580; 1:100) and CldU tracts with a rat anti-BrdU/CldU antibody from Eurobio (ABC117–7513; 1:100). The following secondary antibodies were used: Alexa fluor 488 anti-mouse antibody (Life A21241; 1:100) and Cy3 anti-rat antibody (Jackson ImmunoResearch 712-166-153; 1:100). Fibers were visualized and imaged by Carl Zeiss Axio Imager Apotome using ×40 Plan Apo 1.4 NA oil immersion objective and acquired using Zeiss Zen 3.1 software. Replication tract lengths were analyzed using ImageJ software.

### DNA combing assay

DNA combing experiments were performed using a previously reported protocol[66] with the following modifications: Cells were plated at 2 × 10⁶ cells per 100 mm dish and allowed to adhere for 24 h. Subsequently, DNA was labeled for 30 min with 100 μM IdU (Sigma-Aldrich) and washed 2× with PBS followed by incubation with 100 μM CldU (Sigma-Aldrich) with or without treatment with replication stress drug/Hydroxyurea (Sigma-Aldrich), depending on the assay. For the fork restraint assays, cells were exposed simultaneously to 100 μM CldU with 0.5 mM HU for 2 h. After labeling, cells were collected with trypsin, washed with 1× PBS, and resuspended in cold 0.5× trypsin in PBS (45 μl per 100,000 cells). 500 μl of cells were transferred to a new tube, briefly heated at 42 °C, and resuspended

with 500 μl melted 2% agarose type VII (SIGMA) to make the agarose plugs. Plugs were let solidify for 20 min at 4 °C and were then digested with Proteinase K (400 mM EDTA pH 8, 10% Proteinase K, 1% Sarcosyl) at 42 °C overnight. The next day, plugs were washed 3× with TE 1× buffer. TE solution was removed and a solution containing 50 mM MES pH 5.5,100 mM NaCl was added to the plugs that were heated at 68 °C for 20 min. Agarose plugs were then dissolved by adding 2 μl of β-agarase (NEB) and incubated at 42 °C overnight. The following day, dissolved agarose plugs were transferred to the combing machine (Genomic Vision) where DNA was combed onto silane-coated coverslips (Genomic Vision COV-002-RUO) following the manufacturer's specifications. Combed coverslips were baked for 2 h at 60 °C, denatured in denaturation buffer (25 mM NaOH, 200 mM NaCl in H₂O) for 15 min, washed 3× with 1× PBS, and dehydrated by increasing concentration of ethanol 70%, 90%, and 100% each for 5 min. For the IF staining, the coverslips were incubated with BlockAid for 15 min (Life Technologies) at RT followed by the primary anti-IdU and anti-CldU antibodies (1 h, 1:25 anti-mouse Becton Dickinson 347580 for IdU and 1:50 anti-rat Abcam ab6326 for CIdU) and then incubated 1 h with the following secondary antibodies: 1:50 Alexa donkey anti-mouse 488 (Life Technologies ref. 21202), 1:50 Alexa goat anti-rat 555 (Life Technologies ref. A21434) in BlockAid (Thermo Scientific). Slides were air-dried for 5–10 min and were mounted with 7 μl mounting media (80% Glycerol and 20% PBS) and sealed with clear nail polish. Track lengths of the CldU signal (in red) were measured in Fiji[67].

### S1 nuclease DNA combing Assay

As stated above for DNA combing, cells were exposed to 100 μM IdU to label replication forks, followed by 100 μM CIdU with 0.5 mM HU for 2 h or left untreated. Subsequently, cells were permeabilized with 5 ml CSK buffer in 10 cm plates (10 mM PIPES, pH 6.8, 0.1 M NaCl, 0.3 M sucrose, 3 mM MgCl₂, EDTA-free Protease Inhibitor Cocktail (Roche)) at room temperature for 10 min, followed by 1 ml S1 nuclease (20 U/ml) (Thermofisher # 18001016) in S1 buffer (30 mM sodium acetate pH 4.6, 1 mM zinc sulfate, 50 mM NaCl) for 30 min at 37 °C. Finally, cells were collected by scraping, pelleted, and resuspended in PBS (45 μl per 200,000). The following steps were the same as described in the DNA combing assay.

### S1 nuclease DNA combing Assay with Olaparib treatment

Cells were treated with 10 μM Olaparib for 2 h, followed by 30 min of CIdU 100 μM to label replication forks. Subsequently, cells were permeabilized and processed as described above.

### Replication fork restart assay

As stated above, cells were exposed to 100 μM IdU to label replication forks, followed by 4 h 5 mM HU treatment and a second label using 100 μM CIdU. The following steps were the same as described in the DNA combing assay.

### HR assay

HR was performed as described[37]. Briefly, we used a DSB-mediated gene targeting strategy with site-specific TALEN nucleases to quantify HR in cells. DLD1 BRCA2⁻/⁻ cells stably expressing full-length GFP-MBP-BRCA2 and the variants (C315S, S273L, and R3052W) were transfected using AMAXA technology (Lonza) nucleofector kit V (Cat. #VCA-1003) with 3 μg of the promoter-less donor plasmid (AAVS1-2A-mCherry) with or without 1 μg of each AAVS1-TALEN encoding plasmids (TALEN-AAVS1-5′ and TALEN-AAVS1-3′, a kind gift from Dr. Carine Giovannangeli). The day after transfection the media was changed and 48 h post-transfection the cells were trypsinized and reseeded on a 10-cm culture dish and cultured for additional 8 days. The percentage of mCherry positive cells was analyzed on a BD FACSAria III (BD Bioscience) using the FACSDiva software and data

were analyzed with the FlowJo 10.5 software (Tree Star Inc.). Viable and single cells were gated using forward scatter (FSC-A) and side scatters (SSC-S). To separate single cells from the doublets, singlets were selected using FSC-W($y$-axis) plotted against FSC-A($x$-axis), mCherry positive cells were detected by plotting mCherry-A($y$-axis) against FSC-A($x$-axis).

## Metaphase spreads analysis

Cells ($1 \times 10^5$) were seeded onto six-well plates on coverslips and treated with Hydroxyurea (SIGMA H8627-1G) (0.5 mM 2 h or 5 mM 5 h) and the following day was arrested in metaphase by adding 0.1 μg/ml colcemid (Thermofisher 15212012) for 3 h. A hypotonic shock was performed by incubating the cells with pre-warmed 16% FBS in water for 40 min. Following the hypotonic shock, the cells were fixed by adding 1 volume of methanol–acetic acid (3:1) into one volume of 16% FBS for 15 min at RT, then methanol:acetic acid (3:1) into one volume of water 5 min RT, then methanol–acetic acid (3:1) 30 min RT, and methanol–acetic acid (3:1) for 15 min 4 °C. DNA was stained with 2% Giemsa (Thermo Fisher 10092013) diluted in Gurr buffer (Thermo Fisher 10582013) for 16 min. Chromosomes were acquired either with a Leica DMRB microscope at ×100 magnification and captured with a SONY DXC 930P camera or with a Zeiss Axioskop 2 plus microscope at ×100 magnification and captured with a Leica DMC6200 camera. Chromosomal aberrations were manually counted using Fiji software. Around 50 metaphases were analyzed for cells of each genotype.

## Anaphase bridges analysis

Cells were seeded onto six-well plates on coverslips and synchronized by a double thymidine block. Cells were treated with 2.5 mM thymidine (T1895-1G, Sigma Aldrich) for 17 h, washed once with PBS, and released into normal growth media (RPMI) for 8 h. Cells were treated again with 2.5 mM thymidine for 15 h, washed once with PBS and released into normal growth media for a total of 10 h (Control cells) or treated with HU 0.5 mM for 2 h (H8627-1G, Sigma Aldrich), washed out from the excess of HU with PBS and then released for 8 h (total release 10 h). Cells were then fixed with ice-cold methanol for 15 min at −20 °C, permeabilized with PBS–0.1% Triton (10254583, Fisher Scientific) for 15 min, and blocked with PBS–4% BSA (A4503-50G, Sigma Aldrich) overnight at 4 °C. Coverslips were stained with DAPI (268298, Merck) and mounted onto microscopy slides with Prolong Glass Antifade (P-36982, Thermo Fisher). Anaphase cells were visualized with a Zeiss Axiovert 200 M microscope for DAPI staining with a PCO edge 4.2 bi (PCO) camera and analyzed manually.

## Electrophoretic mobility shift assay

DNA substrates for EMSA were purchased PAGE-purified from MWG Eurofins. The sequence of the oligonucleotides used for these assays is included in Table S1. The ssDNA substrates used in EMSA were oAC379 $^{32}$P labeled at the 5′-end. To generate the 42 bp dsDNA substrate, oAC405 was $^{32}$P labeled at the 5′-end and annealed in a 1:1 molar ratio to oAC406. To generate the 191 bp dsDNA substrate, we used a purified PCR fragment containing the sequence encoding for the human BRC4 fragment of BRCA2 using the plasmid pAC137 (pCMV GFP-MBP-BRC4) and oligonucleotides oAC596 and oAC597. The purified product was dephosphorylated using Antarctic phosphatase (NEB) before $^{32}$P-labeling at the 5′-end. The proteins were incubated at the indicated concentrations with either 5 nM dsDNA (42mer), 6 nM ssDNA ($dT_{40}$), or 1,5 nM dsDNA (191mer) $^{32}$P-labeled DNA substrates for 1 h at 37 °C in EMSA reaction buffer (25 mM Tris Acetate pH 7.5, 1 mM DTT, 1 mM $MgCl_2$, 2 mM $CaCl_2$). The protein–DNA complexes were resolved on 6% native polyacrylamide gels in 1xTAE buffer (40 mM Tris acetate, 0.5 mM EDTA) at 70 V for 75 min. The gels were dried and analyzed with a Typhoon PhosphorImager (Amersham Biosciences) using Image

Quant software (GE Healthcare). In all EMSAs, the ratio of DNA–protein complexes were calculated as the percentage of bound DNA compared with the free DNA.

## DNA strand exchange assay

DNA substrates for strand exchange assay were purchased PAGE-purified from MWG Eurofins. The sequences of the oligonucleotides used for these assays are included in Supplementary Table 1. To generate the radiolabelled dsDNA substrate, oAC1076 was $^{32}$P labeled at the 5′-end and annealed in a 1:1 molar ratio to oAC1077. The 3′ overhang substrate was produced by annealing $^{32}$P-labeled oAC403 (42mer 5′) to oAC423 (167mer) a 1:1 molar ratio. The gapped DNA substrate was produced by annealing oAC423, oAC403, and oAC490 in a 1:1:1 ratio. RPA (100 nM) or storage buffer was pre-incubated with 668 nM (nt+bp) of 3′tail DNA or gapped DNA for 5 min at 37 °C. Then, RAD51 (380 nM) alone or with the indicated concentrations of BRCA2 were added to the mix and incubated for 5 min at 37 °C in a buffer containing 25 mM Tris Acetate pH 8.0, 1 mM DTT, 2 mM ATP, 1 mM $MgCl_2$, 2 mM $CaCl_2$, 0.1 mg/ml BSA (NEB). The reaction was started by adding 4 nM molecules of the donor template dsDNA (oAC1076 and oAC1077 1:1) and the mix was further incubated for 30 min at 37 °C. The reaction was stopped by incubation with 0.25% SDS and 0.5 mg/ml Proteinase K for 10 min. The samples were loaded on a 6% polyacrylamide gel and migrated at 70 V for 75 min. The gels were dried and analyzed with a Typhoon PhosphorImager (Amersham Biosciences) using Image Quant software (GE Healthcare). The percentage of DNA strand exchange product was calculated as labeled product divided by the total labeled input DNA in each lane.

## EdU cell cycle analysis

To label replicated DNA, cells were incubated with 10 μM EdU for 2 h. Samples were collected by trypsinization and incorporated EdU was detected using the Click-iT EdU Alexa Fluor 647 Flow Cytometry Assay Kit (Molecular Probes-Thermo Fisher Scientific) according to manufacturer's instructions. Cells were re-suspended in PBS containing 20 μg ml$^{-1}$ propidium iodide (Sigma) and 10 μg ml$^{-1}$ RNase A (Sigma) before samples were processed using flow cytometry (BD FACSCalibur, BD Biosciences). A number of 10,000 events were analyzed per condition using FlowJo software.

## Statistical analysis

The total number of experimental replicates, mean, median, and error bars are described in the figure legends. Statistical difference was calculated using a two-way ANOVA test with Dunnett's multiple comparisons tests (Fig. 1c–f) or Tukey's test (Fig. 7b and Supplementary Fig. 7). Kruskal–Wallis test followed by Dunn's multiple comparison tests (Figs. 2a, b, 3a, b, 4b, 5ai-b, c, 6b, 7a i, ii, Supplementary Fig. 4a, c, Supplementary Fig. 5). For Fig. 4a, statistical differences were obtained using a one-way ANOVA with Dunnet's test from the mean of four experiments. For Fig. 5aii, statistical differences were obtained using a one-way ANOVA with Tukey's on the difference of the mean CldU track from −HU vs. +HU calculated by unpaired $t$-test with Welch's correction for each cell line, as indicated in the legend. All analyses were conducted using GraphPad Prism version Mac OS X 9.4.0 (453) version.

## Reporting summary

Further information on research design is available in the Nature Portfolio Reporting Summary linked to this article.

# Data availability

The data generated during the current study are included in this published article and its Supplementary information files. Materials are available from the corresponding author on request. Source data are provided with this paper.

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

## Acknowledgements

We thank all members of Carreira lab for their input on this manuscript. We thank Sarah Lambert for helpful discussions on this project. We would like to acknowledge Xavier Veaute (CEA-DRF-IRCM-CIGEx) for the purification of RPA. We thank Laetitia Besse from the Cell and Tissue Imaging Facility of the Institut Curie (PICT), a member of the France BioImaging National Infrastructure (ANR-10-INBS-04), and Charlene Lasgi from the Flow Cytometry platform of Institut Curie, Orsay. This research was funded by grants from the Agence National de Recherche (ANR-17-CE12-0016), Institut National du Cancer (INCa-DGOS_8706), and Agencia Española de Investigacion (MCIN/AEI) (PID2020-115977RB-I00) to A.C. and Jeunes Chercheuses Jeunes Chercheurs from the Agence National de la Recherche (REPLIBLOCK ANR-17-CE12-0034-01) to C.R. A predoctoral Fellowship from Fondation ARC pour la recherche sur le cancer and French Ministry of Education supported D.V. A predoctoral Fellowship from Fondation pour la Recherche Medicale (FRM) supported I.D., A.M., and C.v.N. L.A-A. was funded by an FPI Fellowship from Agencia Española de Investigacion (MCIN/AEI) (PID2020-115977RB-I00). R.C. was funded by a PSL (Paris Science et Lettres) predoctoral Fellowship and R.L. was funded by Fondation ARC pour la recherche sur le cancer.

## Author contributions

D.V., C.M., I.D., C.R., and A.Ca. designed the experiments; D.V., C.M.; I.D., A.M., L.A-A., J.G-E., R.C., R.L., C.v.N., V.B., and C.R. performed the experiments; D.V., C.M.; I.D., A.M., L.A.-A., J.G-E., R.C., R.L., C.v.N., C.R., A.Co., and A.Ca. analyzed the data; A.Ca. wrote the manuscript with input from all the authors and A.Ca. supervised the research. Funding Acquisition, C.R., A.Co., and A.Ca.

## Competing interests

The authors declare no competing interests.
