## [Peer Review File · Nature Communications]

Replication gap suppression depends on the double-strand DNA binding activity of BRCA2REVIEWER COMMENTS

Reviewer #1 (Remarks to the Author):

In this manuscript entitled "Replication gap suppression depends on the double-strand DNA binding activity of BRCA2", Vugic and colleagues investigate how BRCA2 limits ssDNA gaps generated during DNA replication stress. They identify that the dsDNA binding activity of BRCA2 NTD is essential to suppress ssDNA gap formation and analyze the breast cancer variant C315S that impairs the recombination repair of replication-associated lesions. Overall, this is a useful report with a series of supportive evidence. The current set of mutants provide great opportunity for clarifying how fork degradation, HR, and gap repair/suppression relate to therapy response. However, by assessing a range of replication stress related outcomes with distinct agents, it is ultimately challenging to interpret how BRCA2 functional domains relate to the responses studied. For publication, these issues should be addressed as outlined below.

Major:

- 1) BRCA proteins are linked to preventing gaps independent of any exogenous replication stress through roles in lagging strand synthesis as well as preventing gaps following replication stress due to roles in restraining replication. BRCA proteins have also been shown to repair gaps via recombination based repair in post-replication. Given the complexity of gap suppression and repair functions, it is critical to define what particular gap related function is being studied in the manuscript. Here, the manuscript falls short to fully articulate this complexity. This is a critical point as the field tries to sort out the key function underlying therapy response. Indeed, given that both C and N-terminal mutants restrain replication in response to HU, this suggests that replication restraint defects are not related to the gap phenotype found in the N-terminal mutant with respect to HU. Accordingly, in vitro analysis reveals the N-terminal mutant is defective for gap repair. Herein lies the complexity, and the opportunity for understanding. Will this N-terminal mutant have a gap phenotype with PARPi? If so, this mutant which displays no sensitivity to PARPi could demonstrate that gaps are not linked to PARPi sensitivity.
- 2) Gaps have been proposed to operate as independent lesions from DSBs, stalled/collapsed forks or degraded forks. However, this manuscript ultimately links gaps to DSBs. If gaps are ultimately linked to DSBs and if the latter drives genomic instability, then one would expect that a BRCA2 mutant that fails to suppress/repair gaps would only be an issue if the BRCA HR function was also disrupted. However, the BRCA2-C315S mutant that fails to suppress gaps is proficient in DSBR. This is an interesting finding that gap suppression and HR are separable. But this finding confounds the model that gaps become DSBs because again why would cells be sensitive if they can repair DSBs. In this context, they argue that this mutant is only disrupted for repair of replication associated DSBs. This is problematic because the lack of PARPi sensitivity could be argued to show that the mutant is capable of repairing replication associated DSBs given that PARPi is routinely proposed to lead to replication associated DSBs. Overall, these issues make the current manuscript difficult to interpret.
- 3) It is highly recommended that underlying phenotypes that relate to PARPi, MMC vs HU are considered. As described above, with the N-terminal mutant that is resistant to PARPi, does PARPi treatment not induce gaps? With the C-terminal mutant that is sensitive to PARPi, does PARPi treatment induce gaps whereas HU does not? Will MMC be mixed? How does the fork degradation, HR and RAD51 localization, genomic instability phenotypes relate to the agent sensitivity or lack of it?
- 4) In Fig. 2A and following figures, the authors start to use only one clone (A7) for C315S. They also give up the investigation of S273L VUS. Please provide reasons or complete related experiments for the other clones and variants.
- 5) The study lacks important controls: i) In Fig. 5B, do gaps form in untreated conditions? ii) In Fig. 7A, for analyzing replication-associated chromosomal aberrations, low dose HU linked to gap formation is not included. iii) In Fig. 7B, metaphase bridges were not analyzed following replication stress.

Minor:

- 1) Since PARPi accelerates replication and induces gaps that latter of which has been proposed to drive synthetic lethality with BRCA deficiency, it is naïve to employ PARPi “as an indirect measure of HR proficiency”. Likewise RS should be more clearly defined.
- 2) The cause of gaps should be clarified. Herein they state, “Replication fork blocks lead to the appearance of stretches of ssDNA or single-strand DNA gaps (ssDNA gaps)”. This statement is a bit misleading or at the very least overly simplistic. As stated, one would envision that uncoupled replication is leading to gaps. However, these gaps form behind the fork and stem from replication either failing to fill lagging strand gaps and/or repriming reactions that ensure the continuation of replication. Thus, there is not a block to replication.
- 3) The references are not entirely accurate and again to enhance clarity it will be important to outline the gap related phenomenon more clearly. For example, while reference 13 describes the source of gaps in more accurate terms that described above, it is not related to BRCA deficiency. In the same issue as ref 13, Cantor DNA Repair 2021 cover this with respect to BRCA proteins.
- 4) In Fig. 1D, the labeling of TALEN is unclear. Please correct them. Also, in Figure S1B, the x-axis FSC-A labeling is cut off. Please adjust.
- 5) In Fig.S4C, the R3052W only showed reduced EdU compared to S3291A. But in line 264-265, the explanation is confusing. Please clarify.
- 6) In line 289-290, the authors said “RAD51 levels at nascent DNA did not decrease in the thymidine chase experiment”. But in Fig. 3B, why the RAD51-EdU PLA foci significantly decreased at 30min HU treatment?
- 7) In Fig. 4A, please add the statistics comparing BRCA2 WT with other variants. Also, should the ratio be CldU/IdU rather than IdU/CldU?
- 8) In Fig. 5A ii), could the authors explain how the difference of the mean CldU track length in -HU vs + HU are quantified? This part is not intuitively obvious based on Fig. 5Ai. Could the fibers after HU be directly compared?
- 9) In Fig. 6A, the pCHK1 is higher in C315S, but why its fiber measurement after HU looks longer than WT in Figs. 5A and 5B?
- 10) In Fig. 7A ii), there is no red bar for the BRCA2-/- HU group.
- 11) The model in Fig. 7C might not be accurate. As shown in Fig. 1D, the BRCA2-C315S cells lacking “dsDNA binding activity” have “nearly normal or intact DSB repair activity by HR”. Thus, decreased HR is not the consequence of loss of dsDNA binding in the figure. Should the HR here be replaced by “GS” or other ssDNA gap repair terms?
- 12) In Table 1, it is unclear to use “+” for responses, add notes or use either sensitive or resistant.
- 13) Some missing words: Line 71: single-stranded “DNA”; Line 119: dsDNA “binding” activity.

Reviewer #2 (Remarks to the Author):

In this manuscript, the authors have examined the role of the N and C terminal DNA binding (NTD and CTD) domains of BRCA2 on the repair of DSB by HR and response to replicative stress. NTD has been previously shown to bind to both dsDNA as well as ssDNA and CTD binds to ssDNA. The authors have used BRCA2 variants that map to these domains to examine their impact on each of the two key functions of BRCA2. R3052W is a known pathogenic variant that is located in the CTD whereas C315S and S273L BRCA2 variants map to the NTD. Based on the sensitivity of the variants to PARPi, MMC, HU and a GFP-based HR reporter, R3051W is considered to be defective in HR but retains normal response to replicative stress. In contrast, C315S and S273L expressing cells are HR proficient but exhibit replicative stress. The authors show that the NTD domain is important for localization of BRCA2 to the nascent DNA but both CTD and NTD are required for recruitment of RAD51 to these nascent DNA at active and stalled replication forks. Cells expressing the CTD as well as NTD variants are defective in protection of stalled forks but exhibit normal restart. Interestingly, a defect in the NTD but not in CTD results in an increase in ssDNA gaps. C315S variant is shown to be defective in RAD51 mediated repair of ssDNA gaps generated by replicative stress. Such unrepaired ssDNA gaps are

converted into DSBs that contributes to an increase in chromosomal aberrations.

Overall, the findings are very interesting as they demonstrate a separation of function between NTD and CTD. The manuscript is very well-written, and the experimental flow is very logical. The only concern is that there are some inconsistencies and some over interpretation of the data, as described below. The authors should address the concerns listed below:

1. The role of NTD in repair of ssDNA gaps is based on the analysis of a single variant, C315S. It is unclear why S273L variant was not included in any of the subsequent experiments. Based on the results shown, it is clear the C315S variant generates ssDNA gaps in response to HU treatment. However, it is a bit premature to conclude that this is the role of NTD. Having similar impact of S273L variants would make it more convincing. The authors should at least examine the repair of ssDNA gaps in cells expressing S273L variants (Figure 4B).
2. Figure 2A: the PLA images for 1 hr. should be included. Also, the images for 30 min. do not represent the data shown in the graph. Based on the images shown, C315S has more PLA foci than S3291A, which is inconsistent with the quantification showing C315S has the least foci.
3. Given the fact that R3052W is defective in both ssDNA and dsDNA binding, how is variant able to bind to the nascent DNA shown in Figure 2A?
4. Figure 3A: PLA foci for 1 hr. should be included to show that C315S has more foci at 1 hr. than BRCA2^{-/-}.
5. Figure 4: The fork protection data shown in A is not convincing at all. The difference between WT and BRCA2^{-/-} is marginal. Although the three variants are not significantly different from BRCA2^{-/-}, they are also not likely to be significantly different from WT (at least C315S). The y-axis should be CldU/IdU, not IdU/CldU.
6. Figure 4B: Why was only C315S variant examined, and other variants were left out of the fork restart assay?
7. Figure S3: Clonogenic survival assay results for R3052Q and C273L C5: there are too few colonies on untreated plates. More cells should have been plated to see their response to MMC and HU. Hard to conclude much based on the results shown.

Reviewer #3 (Remarks to the Author):

The manuscript from Carreira and colleagues is an exploration of the replication and repair phenotypes of mutants of the DNA-binding portions of BRCA2.

The manuscript is very clear in its figures and writing style, the data are of high quality and the work presented supports the findings given. The methodology is sound and reaches the expected standard. The authors extend previous in vitro assays exploring the function of the BRCA2-NTD – showing it can bind to ssDNA gaps and facilitate the exchange of RPA for RAD51 at those gaps. It, and the CTD, contribute to BRCA2 localisation (retention?) at nascent DNA and to RAD51 loading at nascent DNA. The novel findings reported are relatively slim – the work extends current knowledge of the NTD dsDNA binding – showing binding to gapped structures in addition to tailed ones, and showing RAD51 exchange and recombination at those structures promoted via the NTD binding. In cells, for the first time, it shows that the mutant NTD has poor fork protection and gap-suppression, but is HR proficient. A finding likely to be of interest to the field, as it relates to the on-going debate about the relevance of replication fork dynamics to PARP-inhibitor sensitivity, is the finding that C315S-BRCA2, while failing to support gap-suppression (or fork-protection) is nevertheless PARPi resistant. As the field's consensus is coalescing around gap-suppression correlating with PARPi sensitivity, I would like to see this finding highlighted in the abstract.

Some needed data is missing.

An immunoblot to show BRCA2 variant expression is essential.

Minor

Can the authors be clearer on when – in relation to Reversal/protection/restart/after restart, the gap suppression is expected to happen? I accept that experimental dissection may be difficult. An unspoken assumption is that gaps are bound via NTD-RAD51 very close to the replication structure -as BRCA2 is with nascent DNA (but not after it) and presumably fixed via TS within 2 hours (the time the analogue is on for prior to S1 incubation) – presumably this relates to the slow fork progression in the presence of HU (Fig 5)? Or would an assay to examine RAD51 at gaps (behind the fork) in mutant cells be enlightening (thymidine chase PLA experiment Fig 2B) reveal differences?

Statements about PARPi being independent of replication need more careful writing.

Fig 5Aii (A7 – what does this mean?)

More information on the conservation (or otherwise) of the NTD activity would be helpful. Is it a patient VUS or do other patient variants impact dsDNA binding?

I'm curious to know why an MTT assay was done for Olaparib but colony assays for MMC and HU (Fig 1C Vs E and F)?

REVIEWER COMMENTS

Reviewer #1 (Remarks to the Author):

In this manuscript entitled “Replication gap suppression depends on the double-strand DNA binding activity of BRCA2”, Vugic and colleagues investigate how BRCA2 limits ssDNA gaps generated during DNA replication stress. They identify that the dsDNA binding activity of BRCA2 NTD is essential to suppress ssDNA gap formation and analyze the breast cancer variant C315S that impairs the recombination repair of replication-associated lesions. Overall, this is a useful report with a series of supportive evidence. The current set of mutants provide great opportunity for clarifying how fork degradation, HR, and gap repair/suppression relate to therapy response. However, by assessing a range of replication stress related outcomes with distinct agents, it is ultimately challenging to interpret how BRCA2 functional domains relate to the responses studied. For publication, these issues should be addressed as outlined below.

Thank you for your appreciation of this work. We understand that the use of distinct agents might not be ideal to pinpoint how gap suppression relates to therapy response. However, the goal of this work was to determine what was the role of the N-terminal DNA binding domain (NTD) as opposed to the canonical C-terminal DNA binding domain (CTD). We used variants affecting both domains and different DNA damaging agents on purpose to orient us into which function of BRCA2 could be dependent on one or the other domain. In doing so, we noticed the sensitivity to replication stress especially upon HU treatment was mainly affecting the cells bearing NTD variants and therefore we focused on this treatment to characterize the function of the NTD further.

Major:

1) BRCA proteins are linked to preventing gaps independent of any exogenous replication stress through roles in lagging strand synthesis as well as preventing gaps following replication stress due to roles in restraining replication. BRCA proteins have also been shown to repair gaps via recombination based repair in post-replication. Given the complexity of gap suppression and repair functions, it is critical to define what particular gap related function is being studied in the manuscript. Here, the manuscript falls short to fully articulate this complexity. This is a critical point as the field tries to sort out the key function underlying therapy response. Indeed, given that both C and N-terminal mutants restrain replication in response to HU, this suggests that replication restraint defects are not related to the gap phenotype found in the N-terminal mutant with respect to HU. Accordingly, in vitro analysis reveals the N-terminal mutant is defective for gap repair. Herein lies the complexity, and the opportunity for understanding. Will this N-terminal mutant have a gap phenotype with PARPi? If so, this mutant which displays no sensitivity to PARPi could demonstrate that gaps are not linked to PARPi sensitivity.

Although the initial goal of this work was not to determine the function that underlies therapy response we understand this set of variants provide a very nice opportunity to get insights on this topic. We have therefore performed DNA combing experiments to address this question. In particular, we performed DNA combing experiments with cells bearing either a N-terminal variant or the C-terminal variant in the presence of 10 uM PARP1 for 2h followed by a 30 min

pulse of CldU. Then incubated the samples with or without S1 nuclease to reveal the ssDNA gaps. Our results show that BRCA2 deficient cells show increased ssDNA gaps as previously reported. Interestingly, upon PARP inhibition, cells bearing C315S did not present detectable ssDNA gaps, in agreement with their lack of sensitivity to PARPi (Fig. 5c). In contrast, R3052W, that was sensitive to PARPi, did show ssDNA gaps in these conditions. These results agree with the lack of sensitivity of cells bearing C315S to PARPi (Fig.1c). Thus, in our conditions, there seems to be correlation between PARPi-induced ssDNA gaps and PARPi sensitivity, as it has been reported (Cong *et al* Mol Cell 2021). However, our variants cannot separate DSB repair capacity and PARPi sensitivity as impaired DSB repair correlates with PARPi sensitivity.

2) Gaps have been proposed to operate as an independent lesions from DSBs, stalled/collapsed forks or degraded forks. However, this manuscript ultimately links gaps to DSBs. If gaps are ultimately linked to DSBs and if the latter drives genomic instability, than one would expect that a BRCA2 mutant that fails to suppress/repair gaps would only be an issue if the BRCA HR function was also disrupted. However, the BRCA2-C315S mutant that fails to suppress gaps is proficient in DSBR. This is interesting finding that gap suppression and HR are separable. But this finding confounds the model that gaps become DSBs because again why would cells be sensitive if they can repair DSBs. In this contexts, they argue that this mutant is only disrupted for repair of replicaton associated DSBs. This is problematic because the lack of PARPi sensitivity could be argued to show that the mutant is capable of repairing replication associated DSBs given that PARPi is routinely proposed to lead to replication associated DSBs. Overall, these issues make the current manuscript difficult to interpret.

As mentioned above, we have performed experiments and found that PARPi does not induce detectable ssDNA gaps in the case of cells bearing C315S. However, these cells show high sensitivity to HU and mild sensitivity to MMC which are routinely replication stress inducing agents. On the other hand, these cells can repair DSBs in a gene targeting assay. Thus, our hypothesis that these cells are unable to repair replication-related lesions still holds. However, we noticed that we have probably misinterpreted the aberrations observed in our metaphase spreads when considering the literature (see for example N. Danford MiMB 2011) and realized that rather than "breaks" (as stated in the original Fig. 7a), these aberrations should be considered as "gaps" as the chromatids are still held together in the majority of the cases. We have now requantified the images and corrected this mistake in Figure 7a.

As mentioned above, we have also performed these experiments with cells bearing R3052W, which is a known pathogenic variant unable to repair DSBs (see also Guidugli *et al* Can. Res. 2012 and Shimelis *et al* 2017 Can. Res.). R3052W showed strong sensitivity to PARPi; and in this case cells showed ssDNA gaps upon PARPi exposure (Fig. 5c). Thus, the fact that R3052W is PARPi sensitive, display ssDNA gaps in PARPi condition and is DSB-repair deficient suggest that PARPi sensitivity may be linked to both the formation of ssDNA gaps and DSBs even in the cases where fork arrest is not impaired. These results do not rule out the possibility that ssDNA gaps might be sufficient to drive PARPi sensitivity as it has been proposed (Cong *et al* 2021, Panzarino *et al* 2020 Can Res).

We found that C315S (and S273L) cells display ssDNA gaps upon HU (Fig. 5b) but not upon PARPi (Fig. 5c) and on the contrary, R3052W showed ssDNA gaps in PARPi but not in HU conditions. These results suggest that HU and PARPi induce different types of ssDNA gaps. These results also suggest that ssDNA gap repair and the DSB repair function of BRCA2 are

located in two separate regions, ssDNA gap repair being mainly driven by the N-terminal DNA binding domain of BRCA2.

3) It is highly recommended that underlying phenotypes that relate to PARPi, MMC vs HU are considered. As described above, with the N-terminal mutant that is resistant to PARPi, does PARPi treatment not induce gaps? With the C-terminal mutant that is sensitive to PARPi, does PARPi treatment induce gaps whereas HU does not? Will MMC be mixed? How does the fork degradation, HR and RAD51 localization, genomic instability phenotypes relate to the agent sensitivity or lack of it?

As explained above, these different drugs were used as an exploratory measure to determine whether or not variants at the C or N-ter had different sensitivities, after that, only HU was used as this was the stronger difference observed between the two. Nevertheless, as requested by this reviewer, we took the opportunity to test the induction of ssDNA gaps upon PARPi treatment. We found ssDNA gaps in BRCA2 deficient cells upon 10 uM PARPi treatment as previously reported whereas no induction of ssDNA gaps in C315S (PARPi resistant). Remarkably, R3052W (PARPi sensitive) also displayed ssDNA gaps (Fig. 5c). In contrast, HU treatment did induce ssDNA gaps in C315S (and now we have shown also that this is the case in cells bearing S273L, Fig. 5B) but not on cells bearing R3052W (Fig. 5b). We think this result shows that the N-terminal DBD probably through its dsDNA binding is required for gap suppression whereas the CTD is not. These findings also favor the idea of different origins of ssDNA gaps and consequences in the cells by HU versus PARPi as mentioned in the response to point 2.

As for fork degradation and RAD51 localization at the fork, given the rather small phenotype observed in this cell setting in fork degradation already for DLD1 BRCA2 deficient cells (Fig. 4a), and more importantly, the fact that other variants affecting different regions showed similar phenotypes, we do not think pursuing these experiments would clarify things further. Rather, we tried to focus on the conditions where the biggest differences were observed among the C-terminal and N-terminal variants to shed light on how BRCA2 suppresses ssDNA gaps.

4) In Fig. 2A and following figures, the authors start to use only one clone (A7) for C315S. They also give up the investigation of S273L VUS. Please provide reasons or complete related experiments for the other clones and variants.

Given that the sensitivity to different drugs and DSB repair efficiency of cells bearing S273L and C315S in both clones used was similar (Fig. 1c-f) and the fact that is only C315S that alters the dsDNA binding activity exclusively (von Nicolai et al 2016) whereas S273L affects both ssDNA and dsDNA binding (Fig. 1b), we decided to continue the next experiments with C315S as it was the one that would tell us whether the dsDNA binding was specifically required for gap suppression. The reason to focus on clone 7 of C315S was that this clone expresses more protein than the other (Suppl. Fig. 2a) and therefore if we observed a reduction in PLA signal, as it is the case, this could not be attributed to the lower levels of BRCA2 protein in these cells. Moreover, the PLA and DNA combing experiments are especially tedious, costly as they require expensive reagents and multiple controls therefore we tried to minimize the number of samples per experiment as much as possible.

Nevertheless, as requested by reviewer 1, we have now included the experiments performed with one clone of S273L variant in non-treated conditions, 30 min and 1h of HU treatment.

The results are incorporated now in the new graphs of Fig. 2a and Fig. 3a. We have also included representative images at 30 min and 1h of HU treatment.

We have also performed the DNA combing experiments with the variant S273L in HU conditions. Our results show that the difference between the track length in untreated vs HU treated cells is shorter suggesting that this variant renders cells defective at restraining replication (Fig. 5a_{ii}). However, this result difficult to interpret because the CldU track length is already shorter in untreated conditions suggesting slower replication (Fig. 5a_i). These results suggest that the ssDNA binding activity of the N-terminal DNA binding might be required for fork arrest. Importantly, cells bearing S273L also displayed ssDNA gaps at low dose of HU treatment (Fig. 5b). These results reinforce our previous findings strongly pointing to the NTD as an important region for BRCA2 ssDNA gap suppression function.

5) The study lacks important controls: i) In Fig. 5B, do gaps form in untreated conditions? ii) In Fig. 7A, for analyzing replication-associated chromosomal aberrations, low dose HU linked to gap formation is not included. iii) In Fig. 7B, metaphase bridges were not analyzed following replication stress.

We have now included the untreated conditions in DNA combing as requested by reviewer 1. We found that in our cell setting, BRCA2 deficient cells showed ssDNA gaps in non-challenging conditions although to much lower extent than in HU conditions, whereas we could not detect cells bearing C315S or S273L. (Suppl. Fig. 5).

We have also used HU at low dose (0.5 mM 2h HU) in metaphase spreads to quantify the type of aberrations observed in conditions of ssDNA gap formation, as requested. As mentioned above, we have also reclassified the breaks into gaps whenever the two ends of a discontinuity in the staining are aligned within the chromosome, according to the literature (Danford MiMB 2011). Our results show that there are more aberrations and in particular more gaps in C315S cells treated for 2h with 0.5 mM HU compared to the BRCA2 WT cells. These levels are similar to those of the BRCA2 deficient cells. This trend is exacerbated in cells treated for 5h at 5 mM HU (Fig.7a_{ii})

Also as requested, we have performed the anaphase bridges experiment upon treatment with 0.5 mM HU 2h. Our results are shown in Suppl. Fig. 7. Although we found increased number of anaphase bridges in C315S cells vs WT in cells left untreated, the levels were similar in the case of HU-treated cells and only BRCA2^{-/-} displayed different number of DAPI bridges compared to the other two cell lines. Also the HU at this concentration did not induce higher number of anaphase bridges even for the BRCA2 deficient cells. These results suggest that the ssDNA gaps observed in HU conditions in DNA combing and metaphase spreads do not necessarily result in anaphase bridges.

Minor:

1) Since PARPi accelerates replication and induces gaps that latter of which has been proposed to drive synthetic lethality with BRCA deficiency, it is naïve to employ PARPi “as an indirect measure of HR proficiency”. Likewise RS should be more clearly defined.

We appreciate that the understanding of PARPi toxicity has evolved in recent years and we have now removed this sentence.

2) The cause of gaps should be clarified. Herein they state, “Replication fork blocks lead to the appearance of stretches of ssDNA or single-strand DNA gaps (ssDNA gaps)”. This statement is a bit misleading or at the very least overly simplistic. As stated, one would

envision that uncoupled replication is leading to gaps. However, these gaps form behind the fork and stem from replication either failing to fill lagging strand gaps and/or repriming reactions that ensure the continuation of replication. Thus, there is not a block to replication.

3) The references are not entirely accurate and again to enhance clarity it will be important to outline the gap related phenomenon more clearly. For example, while reference 13 describes the source of gaps in more accurate terms that described above, it is not related to BRCA deficiency. In the same issue as ref 13, Cantor DNA Repair 2021 cover this with respect to BRCA proteins.

Point 2 and 3. This statement has been now corrected as such:

Replication stress lead to the appearance of stretches of ssDNA or single-strand DNA gaps (ssDNA gaps). These gaps have been shown to accumulate in BRCA1/2- deficient cells especially under replication compromising conditions such as nucleotide depletion induced by hydroxyurea or after multiple rounds of cisplatin treatments, suggesting the involvement of these factors in preventing replication-associated ssDNA gaps^{8,11,12}. Ref Cantor *et al* 2021 has also been included.

4) In Fig. 1D, the labeling of TALEN is unclear. Please correct them. Also, in Figure S1B, the x-axis FSC-A labeling is cut off. Please adjust.

This labelling refers to the symbols used, open circles (without TALEN), filled circles (with TALEN). We have now added this information to the legend to clarify.

Thanks for spotting this mistake on the FSC-A label being cropped, it is now adjusted.

5) In Fig.S4C, the R3052W only showed reduced EdU compared to S3291A. But in line 264-265, the explanation is confusing. Please clarify.

Thank you, this phrase was confusing and has been removed and the statement is reduced to this: No significant difference was observed in the EdU PLA signal in cells expressing BRCA2 WT compared to C315S, R3052W, or S3291A (Suppl. Fig. 4c).

6) In line 289-290, the authors said "RAD51 levels at nascent DNA did not decrease in the thymidine chase experiment". But in Fig. 3B, why the RAD51-EdU PLA foci significantly decreased at 30min HU treatment?

thank you. This statement has been now corrected to reflect the quantification in Fig. 3B:

RAD51 levels at nascent DNA also decrease in the thymidine chase experiment but to a much lower extent than BRCA2 indicating that RAD51 is bound to the chromatin as well as at the nascent DNA as previously shown^{8,38} (Fig. 3b).

7) In Fig. 4A, please add the statistics comparing BRCA2 WT with other variants. Also, should the ratio be CldU/IdU rather than IdU/CldU?

The statistics comparing the variants with BRCA2 WT has been added. The ratio was mistakenly labeled and has been corrected, thank you. We have also included the mean value on the top of the graph for clarity.

8) In Fig. 5A ii), could the authors explain how the difference of the mean CldU track length in -HU vs + HU are quantified? This part is not intuitively obvious based on Fig. 5Ai.

Thanks for pointing this out. There was a mistake in this panel, it is now corrected in the new version and we included S273L. Quantification: For each cell line, the difference of the

means in the two conditions (untreated vs HU) were obtained from unpaired t-test comparisons. This 'difference of the mean' value and the SEM was then included in a new table for each clone are then statistically compared using a one-way ANOVA.

Could the fibers after HU be directly compared?

We believe the comparison between conditions for each cell line is more accurate than comparing the clones treated with HU directly because the track length starting point is different for each of them as observed in the graph Fig. 5ai.

9) In Fig. 6A, the pCHK1 is higher in C315S, but why its fiber measurement after HU looks longer than WT in Figs. 5A and 5B? This is what we observe. In Fig. 5a and 5b the median size is indeed longer after HU in C315S probably because it contains gaps and there is a mild defect in the restraining of the fork as shown in Fig. 5Aii. The pCHK1 signal being higher might be because of the presence of ssDNA gaps in C315S indicating that the checkpoint activation is working unlike in the BRCA2^{-/-}. There seems to be also slightly more cells in S-phase in C315S compared to WT and BRCA2^{-/-} (although not significant, Fig. 6b) that may contribute to the increased in this signal.

10) In Fig. 7A ii), there is no red bar for the BRCA2^{-/-} HU group.

This has been corrected. Thank you. We have also included the +0.5 mM HU condition.

11) The model in Fig. 7C might not be accurate. As shown in Fig. 1D, the BRCA2-C315S cells lacking “dsDNA binding activity” have “nearly normal or intact DSB repair activity by HR”. Thus, decreased HR is not the consequence of loss of dsDNA binding in the figure. Should the HR here be replaced by “GS” or other ssDNA gap repair terms?

To avoid confusion and more accurately reflect our findings we have substituted the term HR by HR gap filling/repair.

12) In Table 1, it is unclear to use “+” for responses, add notes or use either sensitive or resistant.

Thank you. This has now been modified.

13) Some missing words: Line 71: single-stranded “DNA”; Line 119: dsDNA “binding” activity. This is now corrected, thank you.

Reviewer #2 (Remarks to the Author):

In this manuscript, the authors have examined the role of the N and C terminal DNA binding (NTD and CTD) domains of BRCA2 on the repair of DSB by HR and response to replicative stress. NTD has been previously shown to bind to both dsDNA as well as ssDNA and CTD binds to ssDNA. The authors have used BRCA2 variants that map to these domains to examine their impact on each of the two key functions of BRCA2. R3052W is a known pathogenic variant that is located in the CTD whereas C315S and S273L BRCA2 variants map to the NTD. Based on the sensitivity of the variants to PARPi, MMC, HU and a GFP-based HR reporter, R3051W is considered to be defective in HR but retains normal response to replicative stress. In contrast, C315S and S273L expressing cells are HR proficient but exhibit replicative stress. The authors show that the NTD domain is important for localization of BRCA2 to the nascent DNA but both CTD and NTD are required for recruitment of RAD51 to these nascent DNA at active and stalled replication forks. Cells expressing the CTD as well as NTD variants are defective in

protection of stalled forks but exhibit normal restart. Interestingly, a defect in the NTD but not in CTD results in an increase in ssDNA gaps. C315S variant is shown to be defective in RAD51 mediated repair of ssDNA gaps generated by replicative stress. Such unrepaired ssDNA gaps are converted into DSBs that contributes to an increase in chromosomal aberrations.

Overall, the findings are very interesting as they demonstrate a separation of function between NTD and CTD. The manuscript is very well-written, and the experimental flow is very logical. The only concern is that there are some inconsistencies and some over interpretation of the data, as described below. The authors should address the concerns listed below:

1. The role of NTD in repair of ssDNA gaps is based on the analysis of a single variant, C315S. It is unclear why S273L variant was not included in any of the subsequent experiments. Based on the results shown, it is clear the C315S variant generates ssDNA gaps in response to HU treatment. However, it is a bit premature to conclude that this is the role of NTD. Having similar impact of S273L variants would make it more convincing. The authors should at least examine the repair of ssDNA gaps in cells expressing S273L variants (Figure 4B). As requested by reviewer 2, we have now included PLA experiments showing that S273L variant also reduces the localization of BRCA2 and RAD51 at replication forks (Fig. 2a, 3a). Given that the C315S did not show any defect in fork restart (Fig.4b) we did not expect S273L to show defects either. However, to address reviewer 2 comments we have performed DNA combing experiments in cells expressing the variant S273L in the presence or absence of RS and further incubated with S1 nuclease to reveal the presence of potential ssDNA gaps. We found that, in agreement with C315S, cells bearing S273L show accumulation of ssDNA gaps in the presence of low dose of HU (0.5 mM 2h) (Fig. 5b). This result reinforces the idea that the NTD is required for ssDNA gap suppression.

2. Figure 2A: the PLA images for 1 hr. should be included. Also, the images for 30 min do not represent the data shown in the graph. Based on the images shown, C315S has more PLA foci than S3291A, which is inconsistent with the quantification showing C315S has the least foci.

Thanks for pointing this out. We have now found more representative images and added the ones for S273L as well. We also included the 1h time point for all conditions.

3. Given the fact that R3052W is defective in both ssDNA and dsDNA binding, how is variant able to bind to the nascent DNA shown in Figure 2A?

R3052W is only defective in ssDNA binding through the CTD, it should not be defective in dsDNA binding and ssDNA binding through the NTD.

4. Figure 3A: PLA foci for 1 hr. should be included to show that C315S has more foci at 1 hr. than BRCA2^{-/-}.

Representative images for 1h are now included in both Fig. 2a and 3a as requested.

5. Figure 4: The fork protection data shown in A is not convincing at all. The difference between WT and BRCA2^{-/-} is marginal. Although the three variants are not significantly different from BRCA2^{-/-}, they are also not likely to be significantly different from WT (at least C315S). The y-axis should be CldU/IdU, not IdU/CldU.

The y-axis label has been corrected, thank you.

The results we have plotted come from 4 different experiments in which 100 tracks were analysed for each condition as showed in the superplot in Fig4a. Overall, we have noticed that in this cell setting (DLD1-/- cells), the level of degradation/resection is particularly small, this makes the difference between BRCA2 deficient cells and BRCA2 WT cells already quite limited. We have now included the statistics to show the difference between the BRCA2 WT and the variants and indeed, neither the difference between the BRCA2 deficient cells and the variants is significant nor the BRCA2 WT vs the variants is significant. Nevertheless, looking at the mean value of the 4 experiments that we have included now in the graph, we find that the cells expressing the variants cannot fully complement fork protection to the level of the BRCA2 WT but we cannot distinguish differences between the variants, as we have described in the text.

6. Figure 4B: Why was only C315S variant examined, and other variants were left out of the fork restart assay?

We wanted to test whether C315S given its phenotype in ssDNA gaps was slower in fork restart but this was not the case. Given that BRCA2-/- show no fork restart defect checking the variants that are not sensitive to replication stress seemed unnecessary.

7. Figure S3: Clonogenic survival assay results for R3052Q and C273L C5: there are too few colonies on untreated plates. More cells should have been plated to see their response to MMC and HU. Hard to conclude much based on the results shown.

We have used several seeding densities according to the plating efficiency of each cell line and these experiments are always performed in triplicates. We found that there was a contrasting issue in the images that were used in the Suppl. Fig. 3 that is now corrected. We believe this improvement in the image will make more readable the plates. We have however included all the data in the source data file (Excel sheet) with all the seeded densities used for the quantification of survival for each condition in the different experiments.

Reviewer #3 (Remarks to the Author):

The manuscript from Carreira and colleagues is an exploration of the replication and repair phenotypes of mutants of the DNA-binding portions of BRCA2.

The manuscript is very clear in its figures and writing style, the data are of high quality and the work presented supports the findings given. The methodology is sound and reaches the expected standard.

The authors extend previous in vitro assays exploring the function of the BRCA2-NTD – showing it can bind to ssDNA gaps and facilitate the exchange of RPA for RAD51 at those gaps. It, and the CTD, contribute to BRCA2 localization (retention?) at nascent DNA and to RAD51 loading at nascent DNA.

The novel findings reported are relatively slim – the work extends current knowledge of the NTD dsDNA binding – showing binding to gapped structures in addition to tailed ones, and showing RAD51 exchange and recombination at those structures promoted via the NTD binding. In cells, for the first time, it shows that the mutant NTD has poor fork protection and gap-suppression, but is HR proficient.

A finding likely to be of interest to the field, as it relates to the on-going debate about the relevance of replication fork dynamics to PARP-inhibitor sensitivity, is the finding that C315S-BRCA2, while failing to support gap-suppression (or fork-protection) is nevertheless PARPi resistant. As the field's consensus is coalescing around gap-suppression correlating with

PARPi sensitivity, I would like to see this finding highlighted in the abstract.

Thank you. To get more insight on the PARPi resistance of cells expressing C315S, we have now performed experiments to find out whether cells bearing the variant C315S or R3052W accumulated gaps upon PARPi. Interestingly, PARPi only induced ssDNA gaps in cells bearing R3052W but not in the ones expressing C315S. Thus, consistently with the current model, PARPi-related gap-suppression correlated with PARPi sensitivity in our conditions. However, based on our findings, we believe that HU and PARPi-induced ssDNA gaps may not be the same, the former probably involving 'clean' DNA ends as opposed to 'dirty' DNA ends arising from trapped-PARP1 on DNA. This may in turn lead to different DNA damage tolerance/ gap filling mechanisms explaining why one cell line shows ssDNA gaps in one condition and not in the other and viceversa. We have now included this last part in the Abstract.

Some needed data is missing.

An immunoblot to show BRCA2 variant expression is essential.

This is shown in Suppl. Fig. 2a and Suppl. Fig. 4b

Minor

Can the authors be clearer on when – in relation to Reversal/protection/restart/after restart, the gap suppression is expected to happen? I accept that experimental dissection may be difficult. An unspoken assumption is that gaps are bound via NTD-RAD51 very close to the replication structure -as BRCA2 is with nascent DNA (but not after it) and presumably fixed via TS within 2 hours (the time the analogue is on for prior to S1 incubation) – presumably this relates to the slow fork progression in the presence of HU (Fig 5)? Or would an assay to examine RAD51 at gaps (behind the fork) in mutant cells be enlightening (thymidine chase PLA experiment Fig 2B) reveal differences?

We believe gap suppression involves at least 2 different functions at different time points; one is the prevention by restraining the replication fork which should be happening at the time of the depletion of nucleotides once the cell is hit by the HU treatment (so before reversal). The second would be when the gaps are left behind the forks and need to be protected from further degradation (presumably after reversal) and filled in through different mechanisms which would happen before fork restart.

Given that RAD51 appears both at replication fork and in the chromatin, it would be difficult to detect differences in a Thy chase experiment in PLA setting.

Statements about PARPi being independent of replication need more careful writing.

Fig 5Aii (A7 – what does this mean?) A7 refers to the clone used in this experiment as we have used 2 clones in the survival assays (A7 and C11).

More information on the conservation (or otherwise) of the NTD activity would be helpful. Is it a patient VUS or do other patient variants impact dsDNA binding?

Yes, C315S is a patient VUS recorded in Clinvar:

<https://www.ncbi.nlm.nih.gov/clinvar/variation/VCV000038241.36>. and S273L is another VUS (https://www.ncbi.nlm.nih.gov/clinvar/variation/566428/?new_evidence=true) that also affects dsDNA binding as well as ssDNA binding (Fig. 1b).

Information about the conservation of this region can be found in our earlier work (Nicolai et al. Natu Comm 2016).

We have now included more data with S273L including PLA at replication forks (new Fig. 2a and new Fig. 3A) and DNA combing (new Figure 5a-b and Suppl. Fig. 5).

I'm curious to know why an MTT assay was done for Olaparib but colony assays for MMC and HU (Fig 1C Vs E and F)?

The reason is simple, MTT is much less tedious than clonogenic survival, however, depending on the drug, the difference between BRCA2 WT and BRCA2-deficient cells (our controls) in our cell setting was not sufficient in the MTT assay and we are obliged to use clonogenic survival to get a good differential. This allowed us to assess unequivocally the variants which generally lay in between WT and BRCA2^{-/-}.

REVIEWERS' COMMENTS

Reviewer #1 (Remarks to the Author):

the manuscript is greatly improved, timely, and will be an important addition to the field.
I recommend the following changes:

Abstract:

Perhaps for clarity consider a modification to a sentence in the current abstract to something like this: The dsDNA binding activity in the N- terminal DNA binding domain (NTD) prevents ssDNA gaps upon nucleotide depletion but not upon PARPi treatment suggesting that these drugs trigger gaps by distinct mechanisms and that the NTD uniquely prevents PARPi -induced gaps. This is in essence more clearly articulated in the discussion lines 560 and 561.

Introduction:

The first paper identifying gaps in BRCA deficient cells with respect to therapy and failure to restrain replication was Panzarino et al. 2021 (ref 47) and that should be referenced when topic is introduced.

Same comment above in abstract, this statement could be more clear "These findings suggest that nucleotide depletion and PARPi trigger distinct gap suppression mechanisms" by changing to "These findings suggest that nucleotide depletion and PARPi trigger gaps in a distinct manner".... that therefore requires a distinct BRCA2 function domain.

Results:

The section "BRCA2 dsDNA binding activity is required to limit HU-induced ssDNA gaps but not PARPi-induced ssDNA gaps" fails to describe the PARPi data shown in Figure 5C.

Table 1 is a very important addition that clearly demonstrates that gaps relate to sensitivity when analyzed with respect to the drug used whereas FP (as revealed in the C315S mutant) does not fully align with sensitivity. I recommend a statement to this point being added to the discussion or results.

Discussion:

Line 543, The current construction is a bit misleading and could clarified by adding "Interestingly, despite the (HU-induced) accumulation of ssDNA gaps"

Reviewer #2 (Remarks to the Author):

The authors have satisfactorily addressed all my concerns.

Reviewer #3 (Remarks to the Author):

The authors have made significant improvements to the manuscript. It is a clear and very careful investigation of the contribution that the NTD, and a dsDNA binding domain mutant within it makes to several aspects of BRCA2 function. The data makes the clear point that HU-sensitivity and failed gap suppression on HU treatment correlates with the (in)ability of the NTD to aid RAD51 loading at gaps. Moreover, it shows that PARPi-mediated gaps are different.

The revisions strengthen the conclusions considerably. I believe this paper will be a key report in the field.

please find below our point-by-point response to the reviewers comments in blue.
The corresponding changes in the manuscript text are highlighted in yellow.

REVIEWERS' COMMENTS

Reviewer #1 (Remarks to the Author):

the manuscript is greatly improved, timely, and will be an important addition to the field.
I recommend the following changes:

Abstract:

Perhaps for clarity consider a modification to a sentence in the current abstract to something like this:

The dsDNA binding activity in the N- terminal DNA binding domain (NTD) prevents ssDNA gaps upon nucleotide depletion but not upon PARPi treatment suggesting that these drugs trigger gaps by distinct mechanisms and that the NTD uniquely prevents PARPi -induced gaps. This is in essence more clearly articulated in the discussion lines 560 and 561.

Thank you for all your constructive comments.

We have changed the last paragraph of the Abstract as requested to enhance clarity.

Here is the updated paragraph: "Thus, these findings suggest that nucleotide depletion and PARPi trigger gaps via distinct mechanisms and that the NTD of BRCA2 prevents nucleotide depletion-induced ssDNA gaps".

Introduction:

The first paper identifying gaps in BRCA deficient cells with respect to therapy and failure to restrain replication was Panzarino et al. 2021 (ref 47) and that should be referenced when topic is introduced.

Thanks. This reference has been added.

Same comment above in abstract, this statement could be more clear "These findings suggest that nucleotide depletion and PARPi trigger distinct gap suppression mechanisms" by changing to "These findings suggest that nucleotide depletion and PARPi trigger gaps in a distinct manner".... that therefore requires a distinct BRCA2 function domain.

Thank you, paragraph changed to: "These findings suggest that nucleotide depletion and PARPi trigger gaps in a different manner and therefore require distinct functions for their repair."

Results:

The section "BRCA2 dsDNA binding activity is required to limit HU-induced ssDNA gaps but not PARPi-induced ssDNA gaps" fails to describe the PARPi data shown in Figure 5C.

Thanks a lot, this information was indeed missing. It is now included in the text and highlighted in yellow (page 16).

Table 1 is a very important addition that clearly demonstrates that gaps relate to sensitivity when analyzed with respect to the drug used whereas FP (as revealed in the C315S mutant) does not fully align with sensitivity. I recommend a statement to this point being added to the discussion or results.

In Page 23 there is already a paragraph along those lines:

"Importantly, analysis of the presence of ssDNA gaps in cells treated with PARPi indicated that cells expressing C315S do not accumulate PARPi-induced ssDNA gaps consistent with their resistance to PARPi. In contrast, cells expressing R3052W that were sensitive to PARPi displayed ssDNA gaps under these conditions. These findings reinforce the idea that PARPi-induced ssDNA gaps correlate with PARPi sensitivity as previously reported¹¹." We have also added a phrase at the end of the results section related to Fig. 5c to state the alignment of

PARPi sensitivity and PARP-induced gaps: “These results perfectly correlate with the sensitivity of these cells to PARPi (Fig. 1c).”

We prefer not to add any statement about the predictability of fork protection (FP) for drug resistance because the FP data we have did not allow us to clearly detect a defect in our BRCA2-mutated cells.

Discussion:

Line 543, The current construction is a bit misleading and could be clarified by adding “Interestingly, despite the (HU-induced) accumulation of ssDNA gaps”

Thanks, this has been now added.

Reviewer #2 (Remarks to the Author):

The authors have satisfactorily addressed all my concerns.

Thanks for your constructive comments.

Reviewer #3 (Remarks to the Author):

The authors have made significant improvements to the manuscript. It is a clear and very careful investigation of the contribution that the NTD, and a dsDNA binding domain mutant within it makes to several aspects of BRCA2 function. The data makes the clear point that HU-sensitivity and failed gap suppression on HU treatment correlates with the (in)ability of the NTD to aid RAD51 loading at gaps. Moreover, it shows that PARPi-mediated gaps are different.

The revisions strengthen the conclusions considerably. I believe this paper will be a key report in the field.

Thanks for your constructive comments.